

# Nearshore Tsunami amplitudes across the Maldives archipelago due to worst case seismic scenarios in the Indian Ocean

Shuaib Rasheed [1], Simon C. Warder [1], Yves Plancherel [1], and Matthew D. Piggott [1]

[1]Department of Earth Science and Engineering, Imperial College London, UK

**Correspondence:** Shuaib Rasheed (s.rasheed18@imperial.ac.uk)

**Abstract.**

The Maldives face the threat of tsunamis from a multitude of sources. However, the limited availability of critical data, such as bathymetry (a recurrent problem for many island nations), has meant that the impact of these threats has not been studied at an island scale. Studies of tsunami propagation at the island scale but across multiple Atolls is also a challenging task due to

the large domain and high resolution required for modelling. Here we use a high resolution bathymetry dataset of the Maldives archipelago, and corresponding high numerical model resolution, to carry out a scenario-based tsunami hazard assessment for the entire Maldives archipelago to investigate the potential impact of plausible far-field tsunamis across the Indian Ocean at the island scale. The results indicate that the bathymetry of the Atolls, which are characterized by very steep boundaries offshore, is extremely efficient in absorbing and redirecting incoming tsunami waves. Results also highlight the importance that local

effects have in modulating tsunami amplitude nearshore, including the location of the Atoll in question, the location of a given island within the Atoll, and the distance of that island to the reef, as well as a variety of other factors. We also find that the refraction and diffraction of tsunami waves within individual Atolls contribute to the maximum tsunami amplitude patterns observed across the islands in the Atolls. The findings from this study contribute to a better understanding of tsunamis across complex Atoll systems, and will help decision and policy makers in the Maldives asses the potential impact of tsunamis across

individual islands. An online tool is provided which presents users with a simple interface allowing the wider community to browse the simulation results presented here and assess the potential impact of tsunamis at the local scale.

## 1 Introduction

The 2004 Boxing day tsunami is the single most significant natural disaster to hit the Maldives in nearly a millennium of recorded history. The widespread impact of the tsunami, the loss of human life and the economic impact of the event is un-

precedented in the history of the country, with losses estimated at over 62% of the Gross Domestic Product (GDP), representing the highest relative losses of any affected country (World Bank and System, 2005). The severity of the aftermath highlighted the necessity of developing a better understanding of tsunami propagation in the Indian Ocean (Okal and Synolakis, 2008). Field studies aiming to quantify the amplitudes and inundation levels of the 2004 Boxing Day tsunami across the Maldives revealed that different regions of the Maldives experienced vastly different levels of damage in spite of them having similar



elevation. These data imply that the amplitude and inundation levels across the Maldives depend strongly on the geography of the islands and possibly on the source location of the tsunami (e.g., Fritz et al., 2006; Kan et al., 2007; Fujima et al., 2006).

Historical impacts of tsunamis are not well documented, and since large tsunamis occur with return periods in the hundreds of years and affect very large scales, data from such events have rarely been recorded since oceanic observations began (Synolakis and Bernard, 2006). As indicated by RMSI (2006), even though there is no documented historical record of a tsunami in the

Maldives prior to the December 2004 event, despite a written history spanning nearly a millennia, it is highly probable that historical tsunamis generated from mega-thrust earthquakes in the Indian ocean have impacted the Maldives. Along this line, Kench et al. (2006) also argue that the Maldives have been exposed to multiple tsunamis during the Holocene but it is not possible to identify tsunami events from statigraphic records due to the extremely dynamic geomorphology of coral islands. Other flooding events known to impact the Maldives, for example those due to the periodic swells originating in the Southern

Ocean impacting the South and Central regions of the Maldives, and meteo-tsunamis in the form of local swells (Wadey et al., 2017) are also not well-documented nor well-understood.

It is essential to understand the impact of potential tsunami events across the country at the island-scale to realize the effectiveness of safety measures, embodied by the "safe island concept" in the Maldives(Naseer et al., 2006), and to evaluate protective mitigation measures. High resolution numerical simulations are necessary to fully understand the impact of potential

tsunamis in the Maldives. Particular geographic features of the Maldives archipelago, through their role in modulating complex wave reflection and refraction patterns, have likely resulted in some regions being impacted less and others being impacted more. The ability to understand how local geography affects tsunami propagation within the archipelago represents a step change in disaster risk management for the Maldives. However, accurate island scale tsunami simulations across the Maldives have not yet been carried out because, until recently, bathymetry data of suitable resolution were lacking. Here we employ

a new bathymetry product built specially for the Maldives (Rasheed et al., 2020). This product represents an amalgamation of new satellite-derived bathymetry ground-truthed with field data gathered over the past century (e.g., Gardiner, 1902, 1903; Sewell, 1936; Hass, 1965; Stoddart, 1966). This new product enables high resolution ocean modelling across the archipelago, and is used here to simulate, for the first time, the nearshore impacts of an array of plausible seismic scenarios in the Indian Ocean on the Maldives.

Tsunami propagation has been modelled at single island scale (e.g. Tonini et al. (2014) in the Seychelles and Gelfenbaum et al. (2011); Dilmen et al. (2018) in Tutuila, American Samoa), at single Atoll scale (e.g. Gica (2015) considered Midway Atoll) and at large basin scales (as used in tsunami warning systems). However, hydrodynamic simulations of tsunami propagation across and within large complex Atoll systems, where local reflection and refraction patterns are simulated with sufficient resolution to account for local effects, is rare. Even where hydrodynamic simulations of such domains have been carried out,

e.g. for Tokelau in the South Pacific (Orpin et al., 2016), complex multi-Atoll interactions have not been considered. The Maldives archipelago, consisting of a string of multiple Atolls, each characterized by many islands and bathymetric features, provides an excellent case study to assess tsunami propagation in a complex multi-Atoll system and learn how across-Atoll and within-Atoll effects can influence the local impact of tsunamis.




Below we develop and discuss tsunami simulations for the Maldives designed to investigate the main factors that influence
local variability in the maximum tsunami amplitudes across the islands. The analysis uses the better observed 2004 Boxing Day
tsunami to validate the model. The model is then run accounting for various plausible far-field tsunami scenarios originating
from around the Indian Ocean.

## 2   Methodology

### 2.1   Geological Setting

The Maldives archipelago stretches for approximately 900 km from North to South, and lies in the central region of the Chagos-
Laccadive ridge, located in the central Indian ocean. The archipelago consists of 25 geographic Atolls, further classified as 16
complex Atolls, five oceanic faros and four oceanic platform reefs (Naseer and Hatcher, 2004). The Maldives archipelago
contains the worlds largest Atoll system, with numerous reefs numbering in excess of 2000, representing 5% of the world's
coral reefs (Emerton et al., 2009). Even though there are more than 1200 individual islands in the archipelago, the islands are
very small, with the largest, Gan island in Hadhunmathi Atoll, being only 5 km$^2$ in size. The islands are extremely flat, with
their topography rarely exceeding an elevation of 2 m above mean sea level (ADB, 2020). Approximately 200 of the islands
are inhabited and numerous others are utilized for other purposes, especially for the tourism industry and the development of
luxury resorts. Limited usable exposed land, rapid population growth in the past few decades and competition for land use,
have led to a state of scarcity such that land reclamation is now taking place across the archipelago on an industrial scale (Duvat
and Magnan, 2019). Studies indicate that the coastline of nearly all inhabited islands of the Maldives have been modified in the
past few decades to cater for this growth (ADB, 2020). This in turn has an impact on the flow regime of the Atolls (Rasheed
et al., 2021, 2020).

While it has been hypothesised that the complex bathymetry of the archipelago, shown in Figure 1(b), contributed to at-
tenuated tsunami amplitudes across the archipelago (Fritz et al., 2006), a quantitative test of this hypothesis has never been
attempted. As can be seen in Figure 1, one of the most prominent bathymetric features is the steep depth gradient characterising
the Atolls and the lagoons, giving rise to the 'leaky bucket' concept that is used to describe flow in and out of the Atoll systems
(Betzler et al., 2016).

The Northern Atolls of the Maldives are generally characterized by few exposed islands, separated by numerous channels
with depths of about 30 m, with small individual reefs hosting islands (Riyaz and Suppasri, 2016). The Southern Atolls tend
to be more continuous, comprising of large shallow lagoons with multiple islands and fewer deep channels. Most large islands
of the Maldives have developed on the rims of the Atolls, with the central lagoon of most Atolls devoid of any large islands
(Luthfee, 1995).

Towards the open ocean, the bathymetry rapidly drops from just below sea level to more than 1000 m within 1 km from the
Atoll rim. Even towards the Atoll basins where the depths are much more shallower, this steepness is evident in bathymetry
charts. The bathymetry of the channels between the Atolls varies across the archipelago; in the Maldives inner sea, which
separates the double chain of Atolls in the central regions of the archipelago running north to south, the bathymetry varies from



300 m to more than 500 m, gradually deepening towards the south east. In the Southern region, the depths of the channels between the Atolls are in excess of 1000 m (Luthfee, 1995).

## 2.2 Seismic scenarios

To build confidence in the model, we initially consider the 2004 event for validation against observational data. We employ the fault mechanism defined by Grilli et al. (2007), which discretizes the fault mechanism into five fault plates, as the primary tsunami-generating mechanism. Poisson et al. (2011) studied the event using various source models in numerical simulations with nested models of Sri Lanka and concluded that the fault model described by Grilli et al. (2007) is adequate for hydrodynamic modelling. This mechanism has also been used by a number of other studies to model the impact of the 2004 Boxing

Day tsunami across the Indian Ocean, e.g. in the Seychelles (Tonini et al., 2014).

To test the sensitivity of our results to the tsunami source model, we also employ here a multi-dimensional seismic scenario of the Boxing Day even, as defined by Tanioka et al. (2006), in addition to the simple fault model described by Grilli et al. (2007). The more complex model involves 12 fault plates in the spatial domain and also includes a temporal discretization of each fault plate. Harig et al. (2008) also used the more complex fault model identified in Tanioka et al. (2006) to study

the nearshore impacts in Indonesia and obtained results comparable to field data. By simulating two fault models, we can evaluate the impact of the fault model on the simulated tsunami. These two scenarios are labelled A (the 5-subfault case (Grilli et al., 2007)) and B (the 12-subfault case (Tanioka et al., 2006)). The parameters for each subfault for these two scenarios are summarised in Table 1.

Further, Okal and Synolakis (2008) identified 11 worst case seismic scenarios across the Indian Ocean which are capable of

generating transoceanic tsunamis, akin to the 2004 boxing day event. They carried out tsunami simulations for these scenarios, assessing the large-scale impact across the Indian Ocean. Wijetunge (2012) used these seismic scenarios to asses the resulting tsunami amplitudes at higher resolution along the coast of Sri Lanka. Here we consider eight selected scenarios (out of the 11 available from Okal and Synolakis (2008), summarised in Table 1), to assess the potential impact of some of the worst most-likely tsunamis that could affect the Maldives and to assess the role of across-Atoll and within-Atoll effects. The parameters

for these eight scenarios are summarised in Table 2, categorized according to the region of origin.

1. **South Sumatra Subduction Zone**

   Studies such as RMSI (2006), which provide return periods for various hazards associated with the Maldives, predict return periods of approximately 140 years for tsunamis with amplitudes greater than 2 m in the Maldives that originate from the South Sumatra subduction zone. Given this relatively short return period, we consider all four plausible

worst-case scenarios from Okal and Synolakis (2008), as summarised in Table 2, with probable impact in the Maldives. Scenario 1 represents a repeat of the 24 November 1833 earthquake, which is reported to have generated a small tsunami observed as far west as the Seychelles. Scenario 2 represents the plausible worst-case scenario for the 1833 earthquake with the same focal geometry, but with an extended fault length of 900 km and increased slip of up to 15 m, releasing a total moment equivalent to the 26th December 2004 megathrust event. Scenarios 1a and 2a are modifications of scenar-



| Scenario | $M_0$ (dyn X cm) | $\Phi$ (deg) | $\delta$ (deg) | $\lambda$ (deg) | $L$ (km) | $W$ (km) | $h$ (km) | $\Delta U$ (m) | Source Center Latitude (deg) | Longitude (deg) | t (min) |
|---|---|---|---|---|---|---|---|---|---|---|---|
| A | $7.6\times10^{30}$ | 323 | 12 | 90 | 220 | 130 | 25 | 18 | 94.57 | 3.83 | |
| | | 348 | 12 | 90 | 150 | 130 | 25 | 23 | 93.90 | 5.22 | |
| | | 338 | 12 | 90 | 390 | 120 | 25 | 12 | 93.21 | 7.41 | |
| | | 356 | 12 | 90 | 150 | 95 | 25 | 12 | 92.60 | 9.70 | |
| | | 10 | 12 | 90 | 350 | 95 | 25 | 12 | 92.87 | 11.70 | |
| B | $7.2\times10^{22}$ | 340 | 10 | 90 | 100 | 100 | 10 | 26.1 | 94.57 | 3.83 | 0 |
| | | 340 | 10 | 90 | 160 | 100 | 10 | 0.0 | 95.55 | 2.37 | 1 |
| | | 340 | 10 | 90 | 150 | 90 | 10 | 29.6 | 95.23 | 3.23 | 1 |
| | | 340 | 10 | 90 | 150 | 100 | 10 | 7.3 | 94.25 | 4.45 | 3 |
| | | 340 | 10 | 90 | 150 | 100 | 27 | 10.9 | 93.80 | 5.73 | 3 |
| | | 340 | 10 | 90 | 150 | 100 | 5 | 7.8 | 94.57 | 6.00 | 5 |
| | | 340 | 10 | 90 | 150 | 100 | 22 | 0.0 | 92.93 | 6.87 | 5 |
| | | 340 | 10 | 90 | 150 | 100 | 5 | 12.1 | 93.70 | 7.13 | 7 |
| | | 340 | 10 | 90 | 150 | 100 | 22 | 16.5 | 92.47 | 8.17 | 7 |
| | | 340 | 10 | 90 | 100 | 110 | 10 | 16.6 | 93.23 | 8.43 | 8 |
| | | 340 | 3 | 90 | 150 | 110 | 10 | 7.7 | 92.08 | 10.45 | 10 |
| | | 10 | 17 | 90 | 100 | 110 | 5 | 1.4 | 92.00 | 12.35 | 12 |

**Table 1.** Summary of the seismic scenarios used to simulate the 2004 Indian Ocean tsunami. Fault mechanisms for Scenario A from Grilli et al. (2007) and Scenario B from Tanioka et al. (2006).

ios 1 and 2 which account for the energy accumulated along the fault plane since 1833, taking into account the energy released due to the 2007 Bengkulu earthquake.

## 2. Arakan Subduction Zone

Two plausible fault mechanisms capable of tsunamigenesis with implications in the far-field are simulated for the the Arakan subduction zone (Table 2). Scenario 3 represents the fault mechanism for the 1762 Bangladesh-Myanmar earth-
quakes. Scenario 4 represents a less probable, but plausible fault mechanism accounting for the residual energy from the 2004 megathrust event. Despite the lack of evidence that a large tsunami greater than 2 m in the Maldives could originate from the Arakan Subduction zone due to the bathymetry of the Bay of Bengal (Okal and Synolakis, 2008), we include these scenarios that originate in the Arakan Subduction Zone as RMSI (2006) predicts tsunamis from this region to have the shortest return period with significant impact to the Maldives amongst all the regions of origin considered in this
study.





| Scenario | $M_0$ | $\Phi$ | $\delta$ | $\lambda$ | $L$ | $W$ | $h$ | $\Delta U$ | Source Center | |
|---|---|---|---|---|---|---|---|---|---|---|
| | (dyn X cm) | (deg) | (deg) | (deg) | (km) | (km) | (km) | (m) | Latitude (deg) | Longitude (deg) |
| **Seismic Zone : Southern Sumatra** | | | | | | | | | | |
| 1 | $6.3 \times 10^{29}$ | 322 | 12 | 90 | 550 | 175 | 15 | 13 | 100.85 | -2.05 |
| 1a | $1.7 \times 10^{29}$ | 322 | 12 | 90 | 350 | 175 | 15 | 6 | 100.85 | -2.05 |
| 2 | $1.13 \times 10^{30}$ | 322 | 12 | 90 | 900 | 175 | 15 | 15 | 101.00 | -2.80 |
| 2a | $6.0 \times 10^{29}$ | 322 | 12 | 90 | 900 | 175 | 15 | 8 | 101.00 | -2.80 |
| **Seismic Zone : Arakan** | | | | | | | | | | |
| 3 | $1.5 \times 10^{29}$ | 324 | 20 | 124 | 470 | 100 | 10 | 6.5 | 93.45 | 19.80 |
| 4 | $2.8 \times 10^{29}$ | 20 | 15 | 90 | 470 | 175 | 10 | 7 | 95.29 | 15.39 |
| **Seismic Zone : Makran** | | | | | | | | | | |
| 5 | $1.9 \times 10^{29}$ | 265 | 7 | 90 | 550 | 100 | 15 | 7 | 64.00 | 25.30 |
| 6 | $3.9 \times 10^{29}$ | 265 | 7 | 90 | 550 | 100 | 15 | 7 | 64.00 | 25.30 |
| | | 280 | 7 | 90 | 450 | 150 | 30 | 6 | 59.43 | 25.98 |

**Table 2.** Summary of the eight plausible fault mechanisms across the Indian Ocean proposed by Okal and Synolakis (2008), which we consider in this study.

### 3. Makran Subduction Zone

We consider two fault scenarios originating from the Makran subduction zone. Scenario 5 considers the combined mechanisms of the ruptures in the three most eastern zones of the fault line as per Byrne et al. (1992), which corresponds to the combined fracture zones triggering the the 1851–1864, 1945 and 1765 earthquakes. Scenario 6 further adds the plausible fault zone corresponding to the 1483 earthquake, extending the fault line further west. RMSI (2006) predicts relatively long return periods for tsunamis that impact upon the Maldives that originate from the Makran subduction zone, with a return period just over 250 years for a tsunami with amplitudes of 2 m in the Maldives.

### 2.3 Numerical Simulations

In this study we use Thetis, a 2D (Angeloudis et al., 2018) and 3D (Kärnä et al., 2018; Pan et al., 2019) flow solver, built using the Firedrake finite element solver platform (Rathgeber et al., 2016) to carry out numerical simulations. Here we use the 2D version of Thetis and solve the depth-averaged shallow water equations in non-conservative form :

$$\frac{\partial \eta}{\partial t} + \nabla \cdot (H_d \mathbf{u}) = 0, \tag{1}$$

$$\frac{\partial \mathbf{u}}{\partial t} + \mathbf{u} \cdot \nabla \mathbf{u} - \nu \nabla^2 \mathbf{u} + f \mathbf{u}^\perp + g \nabla \eta = -\frac{\tau_b}{\rho H_d}, \tag{2}$$





where $\eta$ is the free surface displacement (m), $H_d$ is the total water depth (m), $\mathbf{u}$ is the depth-averaged velocity $(\mathrm{m\,s^{-1}})$,
comprising the $x$- and $y$-components $u$ and $v$, and $\nu$ is the kinematic viscosity of the fluid $\mathrm{m\,s^{-1}}$. The term $f\mathbf{u}^{\perp}$ accounts for
the Coriolis force, where $f = 2\Omega\sin(\zeta)$, with $\Omega$ the angular rotation of the Earth, $\zeta$ the latitude and $\mathbf{u}^{\perp}$ the velocity vector
rotated $90°$.

The model uses a discontinuous Galerkin finite element discretization (DG-FEM), using the $\mathrm{P}_1^{\mathrm{DG}}$–$\mathrm{P}_1^{\mathrm{DG}}$ velocity-pressure
finite element pair. For time-stepping, a semi-implicit Crank-Nicolson approach is applied with a constant time step of $\Delta t$. For
all simulations a time step of $\Delta t = 5$ s was used.

The model treats wetting and drying according to the formulation of Kärnä et al. (2011), which introduces a modified
bathymetry $\tilde{h} = h + f(H)$, ensuring a positive total water depth in the model equations, with $f(H)$ defined as:

$$f(H) = \frac{1}{2}(\sqrt{H^2 + \alpha^2} - H), \tag{3}$$

where $H$ is the water height, and $\alpha$ is a tunable constant which for all simulations in this work was set to $\alpha = 2.0$.

Bed shear stress $\tau_b$ is implemented through the Manning's $n$ formulation as

$$\frac{\tau_{\mathrm{b}}}{\rho} = gn^2\frac{|\mathbf{u}|\mathbf{u}}{H_d^{1/3}}. \tag{4}$$

To derive the free surface elevation from the fault parameters, we use the Okada analytical formulae (Okada, 1992) and
assume that the initial instantaneous free surface water displacement is equal to the sea floor displacement and impose these as
either the initial condition or at the appropriate time in the model (table 1).

### 2.3.1 Model Setup

Each scenario was run in a full simulation which spatially extended across the Indian Ocean as seen in Figures 2 (a) and 1 (a),
as well as a nested simulation focused on the Maldives archipelago, as seen in Figures 2 (b) and 1 (b). Absorbing boundary
conditions were set using a sponge region of increased viscocity at the open boundaries of the full simulation. For the nested
simulation, tsunami amplitudes from the full simulation were used to force the model at the domain boundary depending on the
direction of the tsunami wave, i.e. here a one-way coupling approach was employed. Tidal forcing was not taken into account,
as earlier studies of tidal amplitudes (Rasheed et al., 2020), have shown that the tidal regime of the region is very small, with
maximum variations of 1 m from the mean sea level.

### 2.3.2 Bathymetry

We used the GEBCO 2019 global bathymetry data set (Weatherall et al., 2015), as seen in Figure 1 (a), for the large-scale
simulation but a finer bathymetry dataset is essential to run meaningful regional hydrodynamic simulations at the scale of
Atolls and islands. This is particularly important for the complex Atoll systems of the Maldives archipelago where bathymetry
can vary from being at sea level to extreme depths within a small spatial distance. The high resolution bathymetry dataset of
the Maldives of Rasheed et al. (2020) and shown in Figure 1 (b) is therefore used for the nested simulations. The use of this
new bathymetry dataset is essential: field studies by Riyaz et al. (2010) and Riyaz and Suppasri (2016), which analysed various

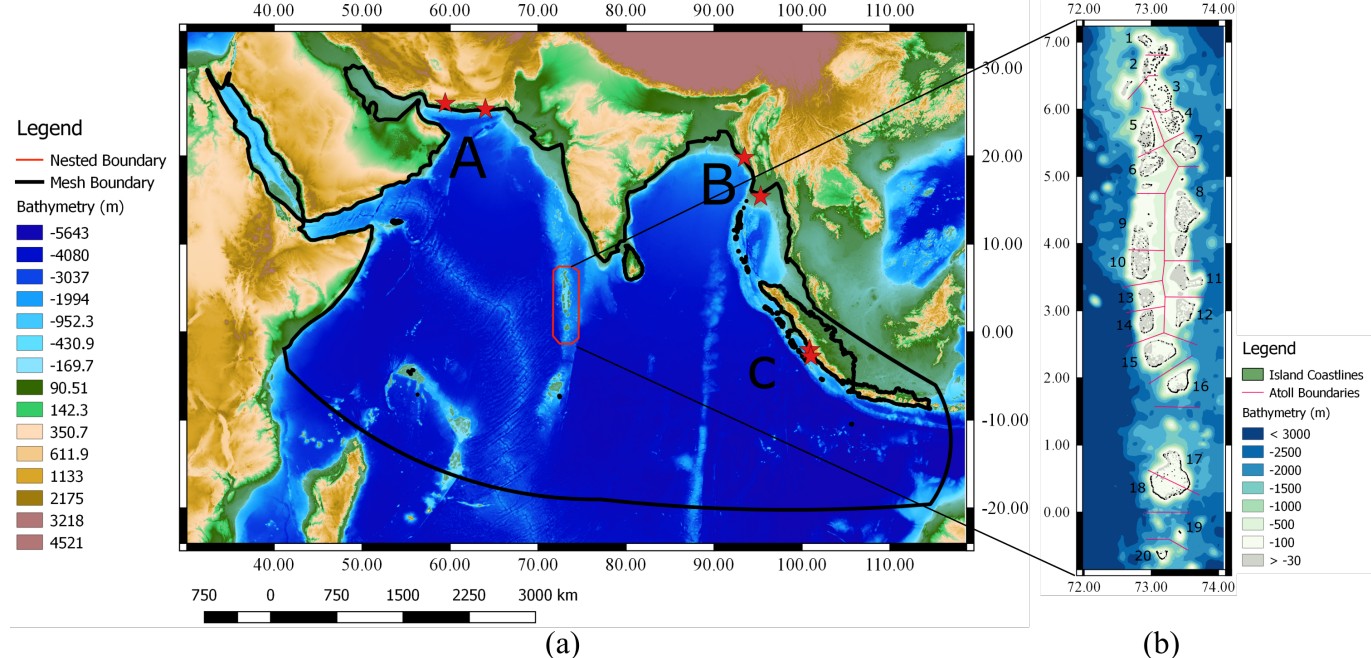

(a)  (b)

**Figure 1.** (a) Bathymetry of the Indian Ocean and topography of the region from GEBCO 2019 (Weatherall et al., 2015) and (b) the bathymetry of the Maldives as per Rasheed et al. (2020) used for the nested high resolution modelling. The seismic zones of fault origins (marked by red stars) are labelled in (a) as A: Makran Subduction Zone, B: Arakan Subduction Zone and B: South Sumatra Subduction Zone. The current administrative Atoll boundaries of of the Maldives archipelago is shown in (b) and labelled using official names as 1. North Thiladhumathi, 2. South Thiladhunmathi, 3. North Miladhunmadulu, 4. South Miladhunmadulu, 5. North Maalhosmadulu, 6. South Maalhosmadulu, 7. Faadhippolhu, 8. Male', 9. North Ari, 10. South Ari, 11. Felidhe', 12. Mulaku, 13. orth Nilandhe, 14. South Nilandhe, 15. Kolhumadulu, 16. Hadhunmathi, 17. North Huvadhoo, 18. South Huvadhoo, 19. Fuvahmulah and 20. ddu Atoll.

statistical correlations between island size, shape, distance to reef edge with respect to the impact of the 2004 Boxing Day tsunami across the Maldives, suggested that small scale features present within the Atolls likely affect wave propagation and therefore tsunami impact. Fujima et al. (2006) carried out a numerical simulation of the 2004 tsunami event using a bathymery dataset with a spatial resolution of 2 arcminutes and highlighted the need for more accurate bathymetry data to account for the complex bathymetry of the Atolls. The bathymetry dataset used here has been used previously for hydrodynamic modelling

of the Maldives at the Atoll (Rasheed et al., 2020) and archipelago-scale (Rasheed et al., 2021), and has been successfully validated against qualitative and quantitative field data.

### 2.3.3   Coastline data

The simulation domain contains a wide variety of coastlines, ranging from numerous small islets in the range of 10 s of metres to continental shorelines in the range of 1000 s of kilometres. Following the study by Rasheed et al. (2021), we used the GADM

database (Areas, 2018), as illustrated in Figure 1 (b), to represent the coastlines of the islands of the Maldives in the nested


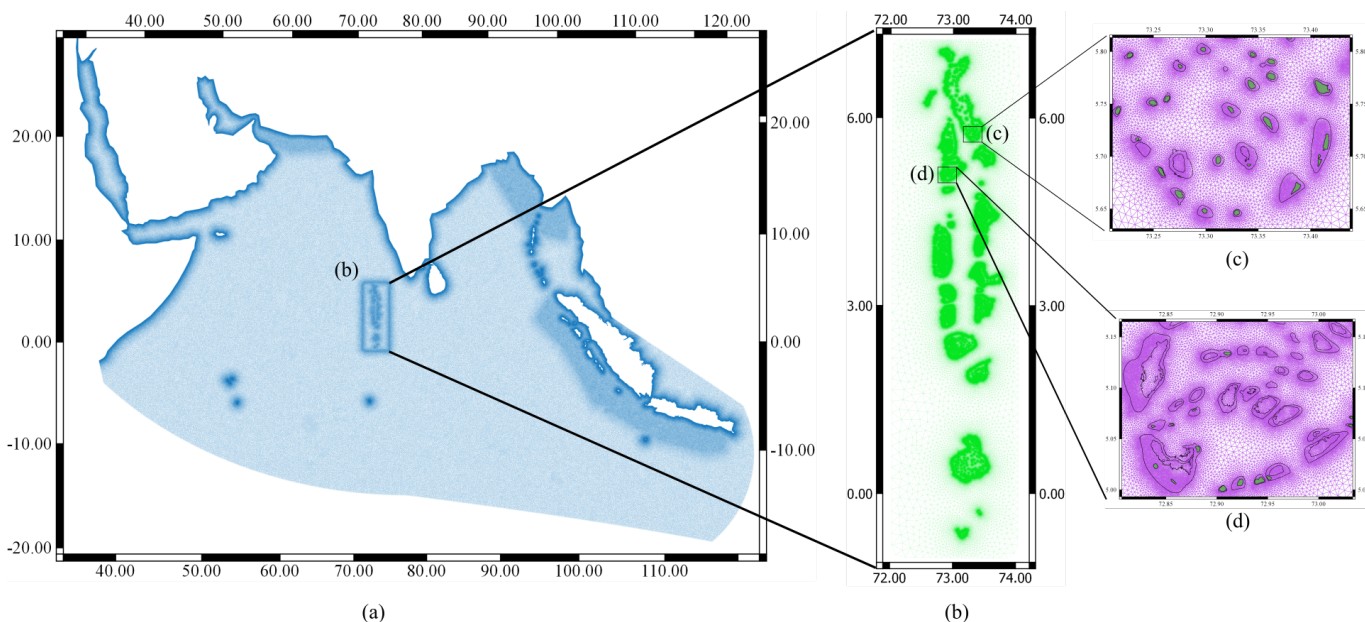

**Figure 2.** (a) The unstructured mesh used for the full simulation, showing increased mesh resolution around fault zones, and coastal areas. (b) The unstructured mesh used for the nested simulations showing increased resolution near the Maldives archipelago. Panel (c) highlights discontinuous reefs of South Midhunmadulu Atoll while panel (d) zooms in on the unique faro system of South Maalhosmadulu Atoll.

simulation and no islands or islets were removed. For the full simulation, we used the "low" resolution GSHHG coastline data from Wessel and Smith (2013), Figure 1 (a), from which we removed smaller islands as the emphasis of this study is the impact of potential tsunamis across the islands of the Maldives archipelago and large-scale propagation is not affected by the presence of small scale islands.

### 2.3.4  Bed friction

A uniform Manning friction parameter of $0.025 \ \mathrm{sm}^{-1/3}$ was applied across the domain for both the full and the nested simulations, and the minimum water depth was set as 0.1 m, applying a quadratic drag coefficient ($C_D$) that varies with the bathymetry, between $\sim 0.0005$ in the open ocean to $\sim 0.026$ in shallow areas. These values are in line with Kraines et al. (1998), who reported similar values in shallow water lagoons and reefs characteristic of the Maldives. Dilmen et al. (2018),
studied the impact of bed friction on tsunami amplitudes in a reef environment and found that the use of different bed friction values yield varying results within different geographic areas in a reef environment. Here, we use these generic values, as bed friction for coral Atolls such as that of the Maldives are not sufficiently known (Rosman and Hench, 2011).



### 2.3.5 Unstructured mesh generation

The Python package *qmesh* (Avdis et al., 2016, 2018) was used to generate the meshes used for the simulations within this
work. The mesh for the full simulation, shown in Figure 2 (a), contains 444,865 nodes and 889,852 triangular elements. The
meshes were generated in the UTM43N coordinate reference system and the mesh element size was set to 15 km in the open
regions and a refinement of 2500 m was applied towards continental coastlines and large islands. A further refinement of 5000
m and 7500 m was set at the Atoll boundaries of the Maldives archipelago and fault regions. Very small features such as small
islands were excluded from the full simulation mesh, while all islands of the Maldives were included in the nested simulation
(see section 2.3.3).

For the higher resolution nested simulation covering the Maldives area, we use a high resolution mesh which contains
1,156,656 nodes and 2,314,856 triangular elements. The mesh element size varied from 10 km at the open boundaries to 50 m
at the coastline of the islands of the Maldives. Further refinement of 100 m was applied towards the shallow lagoon areas as
shown in Figure 2 (b1) and (b2). These mesh refinement scales were selected based on the mesh resolution study by Rasheed
et al. (2021), which showed that this scale of refinement is necessary to fully capture the numerous small scale features of the
complex Atolls of the Maldives.

## 3 Results

### 3.1 Simulations of the 2004 Boxing Day tsunami

The Indian ocean tsunami of 2004 provides observational data, which can be used for model validation purposes. Various
investigators have gathered field data across the Maldives archipelago documenting the aftermath of the 2004 tsunami. Since
we here use both qualitative and quantitative data from these sources, we first present a brief summary of surveys carried out in
the Maldives that report tsunami amplitudes at different points across the archipelago and then compare results from the 2004
Boxing Day simulations with these field observations.

### 3.1.1 Observed tsunami amplitudes from field surveys

Fritz et al. (2006) carried out a field survey across 11 islands in the central Maldives, which included the islands worst affected
by the 2004 Boxing Day tsunami. The survey noted that the low-lying nature of the islands, likely combined with the local
bathymetry, resulted in complete flooding of the islands surveyed and that classic tsunami features such as different inundation
lines at different elevations and tsunami run-up could not be observed. The maximum tsunami heights reported in the study are
of order 1 to 4 m but these were noted to be less than half of the values reported in east Africa at almost twice the distance.

Fujima et al. (2006) carried out a similar survey across seven Maldivian Atolls, with similar conclusions to Fritz et al.
(2006). The survey of Fujima et al. (2006) also included locations with tide gauges in the North, South and Central regions
of the archipelago. The survey results of Fujima et al. (2006) highlight the high degree of regional variability of the tsunami
amplitudes along the coastlines and between islands, even if the islands show little variation in topography. For instance, the





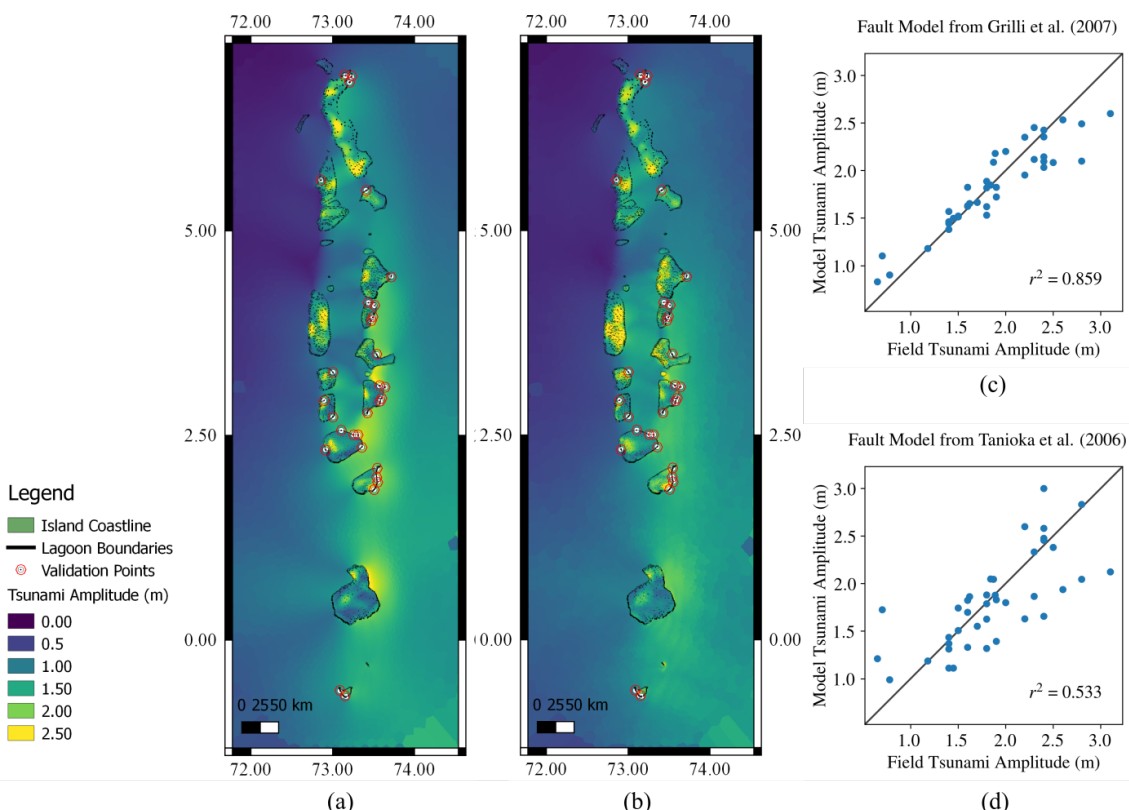

**Figure 3.** Maximum tsunami amplitude across the Maldives, modelled using (a) the fault model described in Grilli et al. (2007) (scenario A) and (b) a multi-scale fault model described in Tanioka et al. (2006) (scenario B). Panels (c) and (d) compare simulated tsunami amplitudes with observations from field surveys for 41 validation points, for scenarios A and B, respectively.

tide gauge in Gan Island, Seenu Atoll, recorded tsunami amplitudes of less than 1 m but survey results from the field study

235    revealed that tsunami amplitudes across the island in the vicinity of the tide gauge varied by more than 1 m.

Kan et al. (2007) carried out an extensive survey across 43 islands of the archipelago to observe the impact of the tsunami, focusing on islands along the eastern rim of the archipelago, where the initial tsunami surge hit the islands. In addition to inundation heights, the Kan et al. (2007) survey also reports the cross-sectional topography of the islands for the locations surveyed, providing a snapshot of the tsunami wave in the local vicinity of the islands and the impact of local topography on

240    inundation across the islands.

Survey results from all of these field studies are included here, however, where the spatial distance is very small, we considered the measurement which was closest to the coastal line, since the model used here is capable to propagate up to the coastline (explicit simulations of inland flooding would require an even higher resolution representation of land feature and additional dynamics).


### 3.1.2 Comparison of the model simulations with field observations

Comparison of model results with field data indicates that the results using the fault model of Grilli et al. (2007) and the more complex fault model of Tanioka et al. (2006) are consistent in predicting the regions of maximum impact across the archipelago (Figure 3 (a) and (b)). Both simulations predict large variations in tsunami amplitude across very small spatial distances, in line with field observations, and consistent with expectations given the complex nature of the bathymetry and coastline of the islands, which is for the first time incorporated in these high-resolution results.

To quantitatively compare the performance of the two models, the Pearson correlation coefficient defined as:

$$r = \frac{\sum_{n=1}^{N}(O_n - \bar{O})(M_n - \bar{M})}{\sqrt{\sum_{n=1}^{N}(O_n - \bar{O})^2 \sum_{n=1}^{N}(M_n - \bar{M})^2}}, \tag{5}$$

where $N$ is the number of data points, $O_n$ and $M_n$ are the observed and modelled values and $\bar{O}$ and $\bar{M}$ are the means of the observed and modelled values was used to quantify agreement in tsunami heights between the two simulations for the 2004 event at 41 locations across the country, where field data is available. Comparison results, in figure 3 (c) and (d), shows that a Pearson correlation coefficient of 0.9 was obtained for the square of the correlation coefficient ($r^2$) for scenario A (using the fault model by Grilli et al. (2007)) while the corresponding value obtained for scenario B (using the fault model by Tanioka et al. (2006)) is 0.73, showing that the maximum tsunami elevations are highly sensitive to fault parameters.

We also note that the islands of south eastern Kolhumadulu Atoll, north eastern Mulaku Atoll and the eastern Hadhunmathi Atoll are shown by the model to experience the maximum impact (Figure 3). Kan et al. (2007) reports that the islands in this zone experienced the maximum damage across the archipelago, observations that are consistent with the model outputs obtained here. Out of the 106 persons either missing or known to have died across the archipelago, this region alone accounted for at least 78. The islands on the north eastern zone of Huvadhoo Atoll (Figure 4 (a) and (e)) were both inhabited and also reported fatalities; the model simulations also predict high tsunami amplitudes there. The south eastern zone of South Nilande Atoll as well as the inner Atoll basin regions of several Atolls, such as North Maalhos Madulu and South Nilande Atoll, are also predicted by the model to have high maximum tsunami amplitudes.

### 3.1.3 Difference in local tsunami amplitude due to two different faulting mechanisms

With a distance of over 2000 km from the fault origin, the Maldives qualifies as far-field for the 2004 Indian ocean tsunami and the use of different fault models is generally not expected to result in significant differences (Okal and Synolakis, 2008). However, we observe differences in maximum tsunami amplitudes for the simulations when the fault model is changed as seen in (Figure 3 and Figure 4). These differences arise from differences in the seismic moment and fault geometry inherent to the two fault models (see Table 1).

This is in line with results from Poisson et al. (2011), who also found differences in modelled tsunami amplitudes across the coast of Sri Lanka when different fault mechanisms were used for the 2004 event. Figure 4 (a–h) shows maximum tsunami amplitudes simulated across various locations for both fault models, indicating that the more energetic fault model by Grilli et al. (2007) leads to a general increase of maximum tsunami amplitude across the domain. However, the maximum tsunami




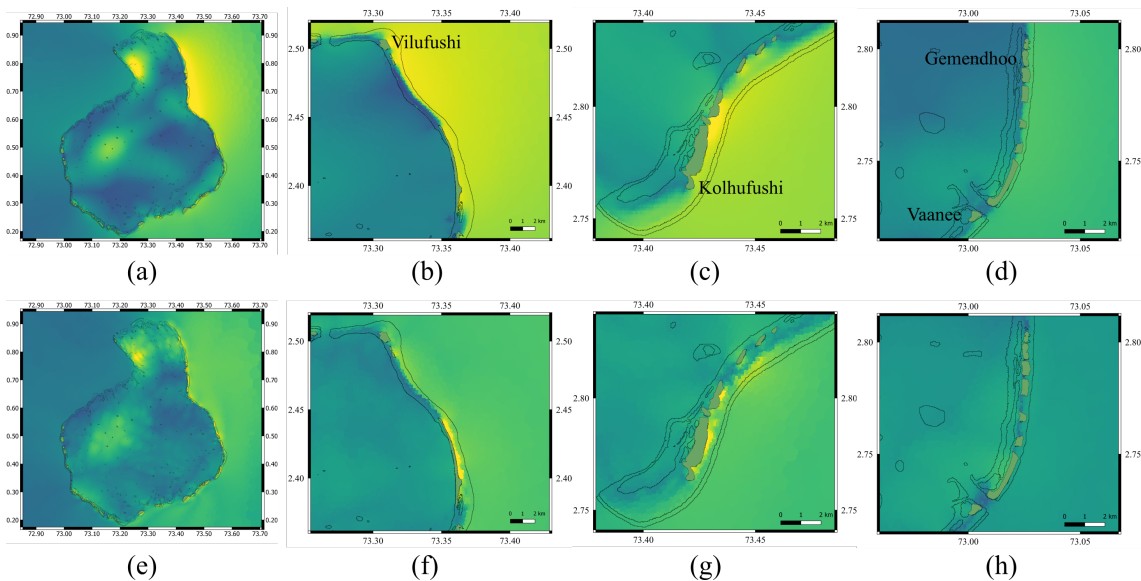

**Figure 4.** Maximum Tsunami amplitude across (a) Huvadhoo Atoll, and the region around the islands of (b) Villufushi (North East Kolhu-madulu), (c) Kolhufushi (south eastern Mulaku Atoll), (d) Vaanee and Gemendhoo (south eastern zone of South Nilande Atoll), using the fault model by Grilli et al. (2007) and (e–h) using fault model by Tanioka et al. (2006).

amplitudes simulated with the fault model by Tanioka et al. (2006) predicts higher amplitudes in some areas. Figure 4 (b) and (c), showing the maximum tsunami amplitude in the vicinity of the hard-hit Kolhufushi island (south eastern zone of Mulaku Atoll), illustrate this phenomenon.

280    The fault models differ in their direction of focus. Figure 4 (b) shows significant increase in amplitude in the north eastern part of Kolhumadulu Atoll, an area which suffered devastation during the event, destroying the inhabited island of Villufushi and significantly altering the geomorphology of the uninhabited island of Kalhufahalaafushi to the South. However, figure 4 (f), shows that the fault model of Tanioka et al. (2006) focuses the maximum impact across the island of Kalhufahalaafushi, leading to no significant increase in tsunami amplitude simulated for Villufushi island.

285  **3.2   Impact of various tsunami scenarios from around the Indian Ocean on the Maldivian archipelago**

   1. **Southern Sumatra Subduction Zone**

      We find that the impact across the Maldives archipelago for the worst case tsunami scenarios simulated with origin in the South Sumatra subduction zone is significantly less in comparison to the 2004 event. However, worst case scenarios of South Sumatra origin can produce tsunamis which can have an impact in the Maldives. The lower severity of waves
290      produced by earthquakes along the Southern Sumatra subduction zone is due to the tsunami wave direction which, in this case, is focused away from the Maldives and towards the south western Indian Ocean (Okal and Synolakis, 2008).

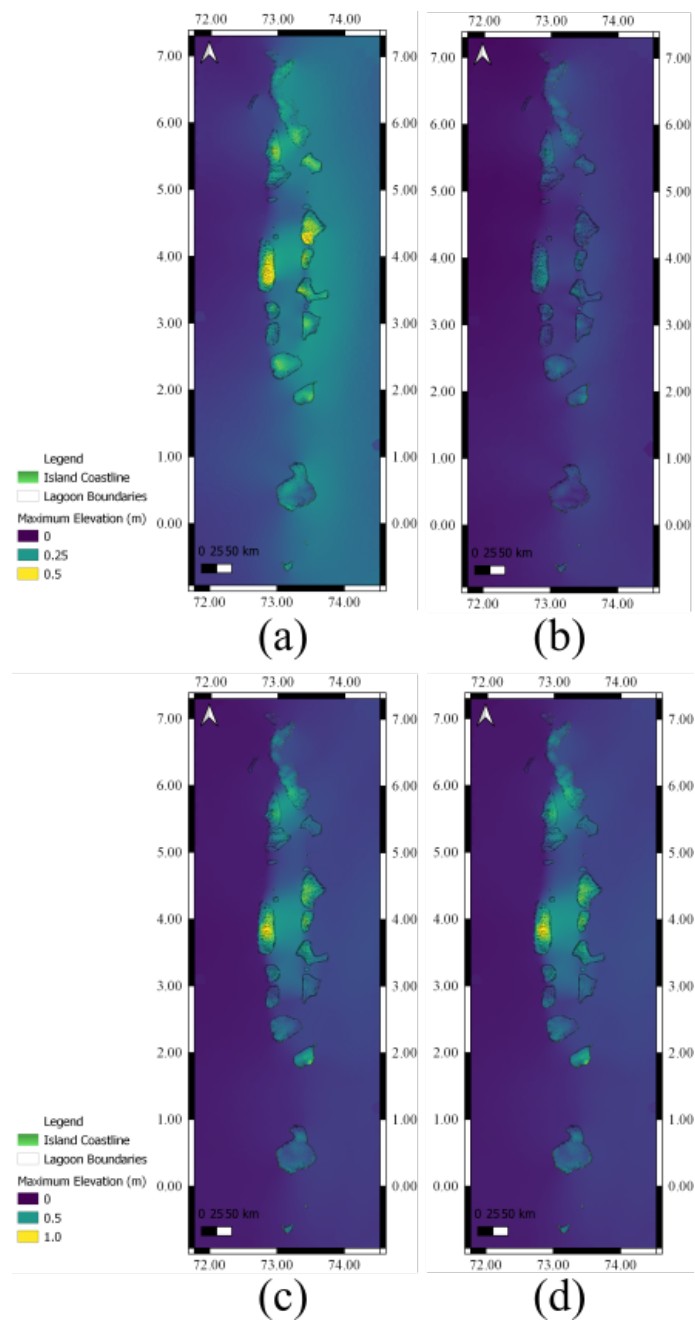

**Figure 5.** Maximum tsunami amplitude across the Maldives for (a) scenario 1, (b) scenario 1b, (c) scenario 2 and (d) scenario 2b from Table 2. Even for the worst case scenarios the maximum tsunami amplitude is relatively small.

Scenario 1 (Figure 5 (a)) is a tsunami simulation of the 1833 South Sumatra earthquake. This event is predicted to have little impact for the Maldives, with maximum local tsunami amplitudes averaging only 0.5 m (Figure 5 (a)). South Male





Atoll, central regions of North Maalhosmadulu Atoll and Ari Atoll are predicted to suffer the greatest impact, but even
then the predicted amplitude is relatively small and there exist no records of damage in the Maldives for that 1833 event.
Scenario 1(a), which simulates the more recent Bengukulu earthquake (2007) also produces very small amplitudes across
the Maldives (Figure 5 (b)) and warrants no further discussion.

Scenario 2, which represents a plausible extension of scenario 1, leads to larger impacts across the archipelago (Figure
5 (c)), particularly for Atolls located in the central region, with tsunami amplitudes close to or exceeding 1 m at some
locations, due to the tsunami being directed towards this region. Given that the maximum tsunami amplitude distribution
simulated for that event is similar to the 2004 Indian ocean tsunami, one would expect that tsunamis generated by an
event similar to scenario 2 could lead to significant damage. Scenario 2a, which is a modification of scenario 2
that reduces the total energy of the tsunami, accounting for the 2007 Bengkulu earthquake, leads to reduced tsunami
amplitude predictions across the Maldives as expected, but the forecasted amplitudes in the Maldives region are still
higher than those predicted for scenario 1. If combined with a high tide, tsunamis generated by scenario 2a would likely
have an impact across locations predicted to have higher amplitudes.

2. **Arakan Subduction Zone**

Two fault scenarios were considered in the Arakan subduction zone as described in Table 2. Scenario 3 considers the
fault mechanism involved in the 1762 earthquake, and scenario 4 simulates a plausible fault to the North of the 2004
boxing day event. As predicted by Okal and Synolakis (2008), simulated amplitudes for both scenarios are relatively
small across the Maldives (Figure 6). This result is in line with simulations predicting that tsunamis originating from
the Arakan subduction zone do not propagate away from the Bay of Bengal due to the seafloor bathymetry (Okal and
Synolakis, 2008). Given that the maximum tsunami amplitudes of the worst case scenarios in the Arakan subduction
zone are minimal, this suggests that earthquakes from the Arakan subduction zones are not likely to be a significant
threat to the Maldives.

3. **Makran Subduction Zone**

Scenario 5 simulates the 1945 Makran tsunami originating from an earthquake along the Makran subduction zone.
Simulation results from this scenario indicate that tsunamis with this origin can have a substantial impact in the Maldives
(Figure 7 (a)), particularly across the Northern Atolls of the Maldives. We also find that due to the bathymetry of the
Arabian sea, tsunami waves originating there reach the Northern Atolls of the Maldives before reaching regions that
are closer in proximity, such as major cities along the coast of western India. Localised maximum tsunami amplitudes
are predicted to be close to 1 m for some key islands of the Northern Maldives. We conclude that earthquakes from the
Makran subduction zone could cause considerable damage, especially if combined with high tidal conditions.

It is plausible that the damage due to the 1945 event in the Maldives was left undocumented due to the fact that the
most impacted region was mostly uninhabited at the time. However, it is expected that events of this amplitude would
have a considerable impact on infrastructure if repeated today. For example, Hanimaadhoo Island, which today hosts


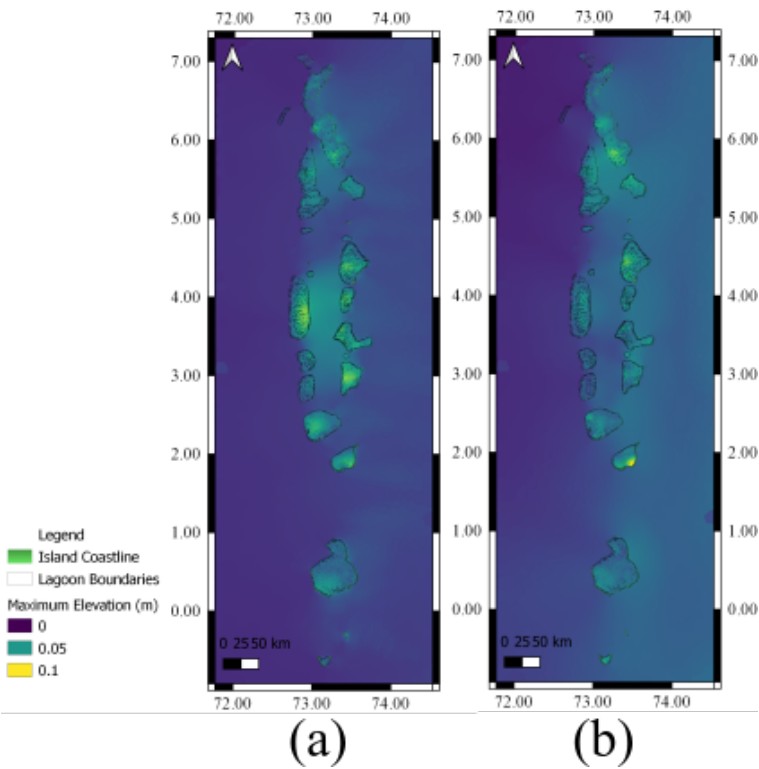

**Figure 6.** Maximum tsunami amplitude across the Maldives due to plausible worst case fault case scenarios in the Arakan Subduction zone, corresponding to scenarios 3 and 4 from Table 2.

the only International Airport in the Northern Maldives, would be expected to face tsunami waves of 0.8 m. Further, the islands of the Northern Atolls of the Maldives are more populous in comparison to the Southern regions of the Maldives. Our results for scenario 5, coupled with recent increases in the populations of some of the impacted areas, indicate that

Makran area earthquakes represent a new risk for the Maldives.

Maximum tsunami amplitudes simulated for scenario 6, which represents an extended fault system in addition to the 1945 earthquake, is predicted to have a slightly smaller impact than scenario 5, in agreement with results by Okal and Synolakis (2008) (Figure 7 (b)). The smaller impact of this larger earthquake generating mechanism is due to destructive interference of the waves near the source.

Even though tsunamis with origin in the Java Subduction zone are predicted to produce waves with implications in the far-field (Okal and Synolakis, 2008), the impact of these tsunamis across the Maldives is predicted to be negligible, in line with similar studies in the region such as Wijetunge (2012) in Sri Lanka.

Further, even though a number of transformation faults exist along the South western Indian Ocean Ridge (SWIO), and some of the world's largest earthquakes have been recorded at these locations, according to Okal and Synolakis (2008),



Natural Hazards
and Earth System
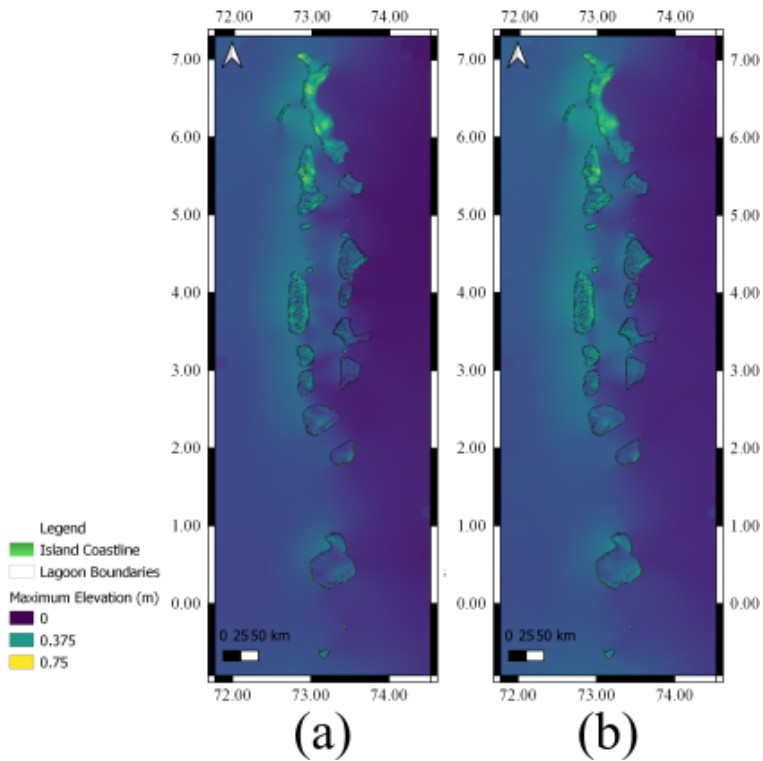

**Figure 7.** Maximum Tsunami amplitude across the Maldives at due to plausible worst case fault case scenarios in the Makran Subduction zone, corresponding to (a) scenario 5 and (b) scenario 6 from Table 2.

none of these are capable of generating tsunamis that impact upon the far-field. Here we exclude Calrseberg Ridge as a plausible fault mechanism on account that RMSI (2006) predicts return periods of ≈400 years for a tsunami with amplitude greater than 1.5 m in the Maldives area.

## 4    Discussion

### 4.1    Importance of local and regional bathymetry-induced refraction/diffraction patterns in modulating the local
impact of the 2004 Boxing Day event

The use of a high resolution bathymetry dataset of the archipelago allows the simulation of tsunami characteristics hitherto not modelled at this scale. Here we summarise and compare some of the observations from the model which demonstrates refraction, diffraction and reflection patterns within the Atolls and show how these contribute to zones of maximum impacts during the 2004 Boxing Day event.

The simulation using the fault model defined by Grilli et al. (2007) are used here to discuss the impact of the 2004 tsunami event in more detail since these yielded better correlation results when comparing data to model results.

Natural Hazards
and Earth System
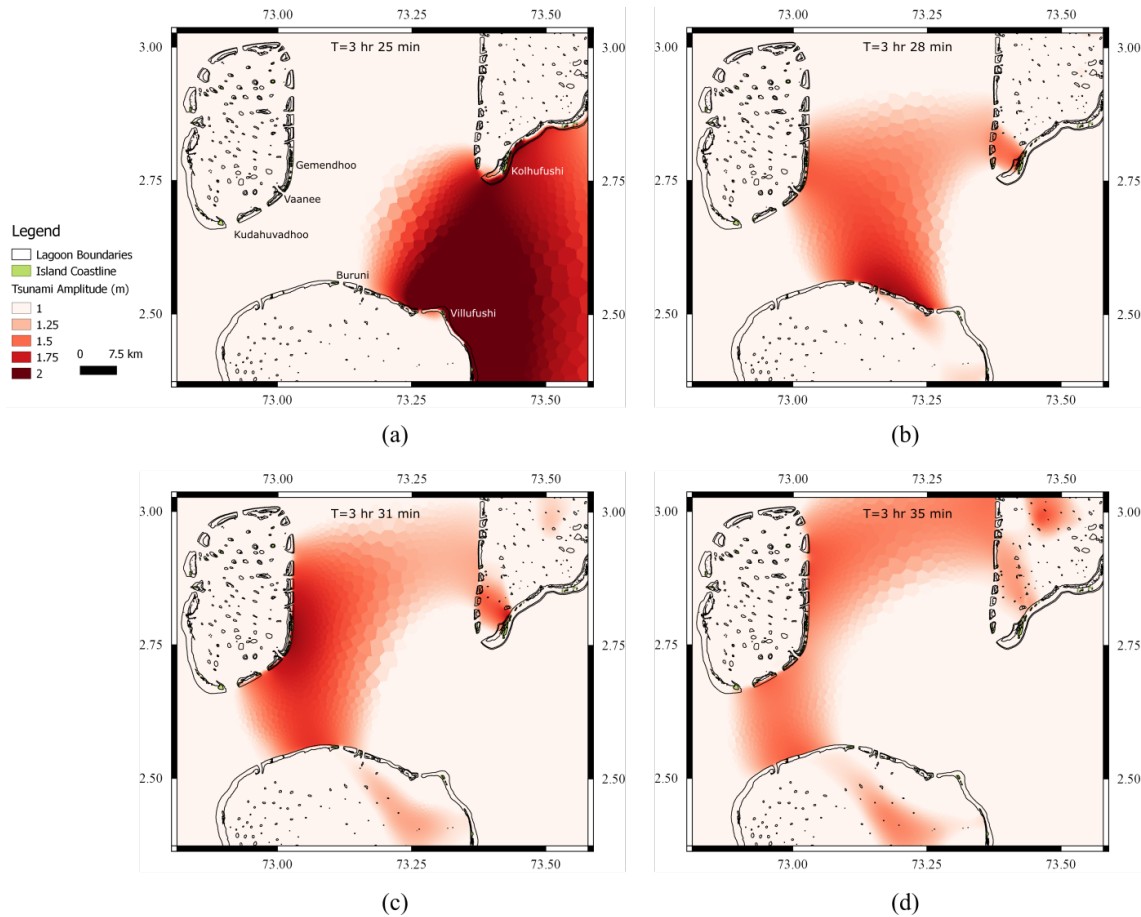

**Figure 8.** The 2004 boxing day Tsunami propagation across *Kudahuvadhoo Kandu* (Sea of Kudahuvadhoo) (the junction of South Nilande Atoll, Mulaku Atoll and Kolhumadulu Atoll) at (a) $T$=3hr 25min, (b) $T$=3hr 28min, (c) $T$=3hr 31min, (d) $T$=3hr 35min. (a) shows the high tsunami amplitude across Kolhufushi island in Mulaku Atoll and Villufushi island in Kolhumadulu Atoll. (b–d) shows the formation of a bow wave across the channel between Mulaku and Kolhumadulu Atolls, which propagates towards Vaanee and Gemendhoo island in south eastern part of South Nilande Atoll.

The steep bathymetry of the Maldivian Atolls and the geormorphology of the lagoons is extremely efficient at reducing the impact of tsunamis at the island scale. This is demonstrated by the fact that, of the various individual scenarios are simulated in this study, none is predicted to have as high an impact across the Maldives as the mega 2004 event, highlighting that the geographic features of the Maldives archipelago naturally attenuate the impact of far-field tsunamis, and extremely large events are required to breach the natural defenses offered by the natural geormorphology of the lagoon, in spite of the low elevation of the islands themselves.

However, the simulations conducted also clearly demonstrate that the location and shape of Atolls plays an important role in amplifying and focusing the tsunami impact locally. The location of Kolhufushi Island in South Mulaku Atoll, combined with





the shape of Kolhufushi lagoon, causes water to build up across the island as seen in Figure 4 (c), (g) and Figure 8 (a). A similar observation is made in Villufushi Island in North Eastern Kolhumadulu Atoll across the channel. The location of the island, in a headland *(Villufusi Muli)* (Luthfee, 1995) projecting out to the area of primary tsunami impact, causes tsunami amplitudes to build up between these islands. These two islands (Kolhufushi and Villufushi), where maximum tsunami amplitudes were recorded across the archipelago, were the only two islands to record double digit figures for people dead or missing.

We also observe that the geometry of north eastern Kolhumadulu and south western Kolhumadulu Atoll forces a surge in tsunami amplitude along the south eastern rim of South Nilande Atoll, as seen in Figure 8 (a)–(d). Both inhabited islands in the region, Gemendhoo and Vaanee, were completely destroyed during the tsunami and remain uninhabited today.

  Model predictions show that while the amplitude of the tsunami is lower at the south western zone of South Nilandhe Atoll in comparison to the eastern rim of Mulaku, Kolhumadulu and Hadhunmathi Atolls, the build-up of the tsunami surge persists

across this region for a longer period of time, due to the very shallow bathymetry of the region with depths less than 3 m. Further, the geormorphology of the region, which consists of a chain of small islets, is predicted to contribute to the build-up of the amplitude with high tsunami flow velocities across the shallow and narrow channels. On the other hand, we also find that Kudauhvadhoo Island in the south of South Nilande Atoll is predicted to have relatively low maximum tsunami amplitudes, in line with field surveys (Hmadhoon Hameed, 2005).

The shape of the Atoll and the location of the island within a large lagoon, as well as the steep bathymetry of the Atoll, caused the build-up of tsunami amplitude to pass through the channel between South Nilande Atoll and Kolhumadulu Atoll without any destructive impact to the island. Residents of both Gemendhoo and Vaanee were evacuated to Kudahuvadhoo on the day of the tsunami by the residents of Kudahuvadhoo and since then have been permanently relocated to the island. These simulations show that local and regional refraction and diffraction patterns can lead to vastly different outcomes, even for

neighboring islands of otherwise similar elevation.

  Furthermore, we find that some of the field observations reported in the literature do not fully agree with other studies of tsunami hazard assessment across the Maldives based on statistical correlations between various parameters such as reef and island geormorphology. To illustrate, Kudahuvadhoo island discussed earlier is frequently predicted by (e.g., Riyaz et al., 2010; Riyaz and Suppasri, 2016) to be very vulnerable to tsunami impact, particularly for the 2004 Boxing Day event, while field

data and results of numerical modelling such as those conducted here, show that the island suffered no visible damage. Hence, we note that statistical correlations between various parameters such as the location of the island from the reef edge alone is not sufficient for tsunami vulnerability mapping for regions such as the Maldives, as these models do not account for the complex bathymetry of the reef environments which can sufficiently alter the local tsunami amplitude. Results of studies at other similar locations, also stress that tsunami wave characteristics are extremely complex. For example, Baba et al. (2008) found that

the Great Barrier Reef reduced the impact of the 2007 Solomon Islands tsunami by several orders of magnitude, while Ford et al. (2014) found that the level of attenuation offered by coral reefs depends on a variety of other factors, which statistical models do not take to account. In the present simulation domain of the Maldives archipelago, we find that the complexity of the tsunami wave propagation is amplified by the presence of numerous coral reefs within individual Atolls, as well as the presence of multiple Atolls within close vicinity.



The use of a high resolution bathymetry dataset allows the simulation to predict features such as an increase in tsunami amplitudes due to shoaling as the tsunami passes through relatively shallow water contributing to increased damage. This is evident in North Maalholmadulu Atoll, where a relatively shallow area with numerous coral reefs and faros is observed to have larger amplitude tsunami waves than surrounding areas. The shoaling effect at Kandholhudhoo Island (the only inhabited island in the zone), combined with a variety of other local factors, such as the absence of coastal vegetation and the removal

of reef for reclamation, lead to the complete destruction of the island; at the time this was the most densely populated island in the Maldives. Similar predictions of higher tsunami amplitudes are made in the shallow water regions of western Ari Atoll and western Kolhumadulu Atoll as seen in figure 3. However, as the islands at these locations were uninhabited at the time of the Boxing Day tsunami, no field data exists for these regions against which to validate the model predictions.

We also observe that the model predicts diffraction and refraction wave patterns inside the Atoll basin, leading to high

amplitudes at islands which are not in the direct path of the tsunami. As seen in figure 3 (a), the central regions of South Nilande Atoll is predicted to have a high tsunami amplitude. This is in agreement with eyewitness accounts and field observations at Rinbudhoo island (Kan et al., 2007), where high inundation depths with the flow of water from all directions was reported. Modelled results agree with these field observations and predict high tsunami amplitudes well within the central basin of the Atoll. Simulation results predict that high tsunami amplitudes at this zone occurred due to the refraction patterns within the

the Atoll basin. Similar observations were made by Ford et al. (2014) based upon simulations of the 2011 Tohoku earthquake in the Majuro Atoll in the Marshall Islands, where tsunami simulations exhibited long lasting tsunami amplitude peaks within the Atoll for up to 60 minutes.

## 4.2    Atoll Tsunami Explorer Tool

Traditionally, the results of tsunami impact studies across the Maldives have been mainly represented via printed charts (e.g.,

RMSI, 2006; ADB, 2020; Riyaz et al., 2010; Riyaz and Suppasri, 2016). However, given the variation in tsunami impact across the archipelago, interpretation of these charts to asses the nearshore impact across the coastline of islands, and especially to assess island scale variations, is extremely difficult. To address this, we here present an interactive application tool built on the Google Earth Engine (Gorelick et al., 2017) that can be used to explore the simulation results presented in this study.

The application, "Atoll Tsunami Explorer", is hosted within the Island Health Explorer (I-Hex) platform (https://i-hex.github.io),

which brings together a host of similar interactive applications specific to the Maldives archipelago based on satellite imagery and various other data sets. An illustration of the user interface of the "Atoll Tsunami Explorer" is shown in Figure 9. Users can explore the maximum tsunami amplitudes at high resolution across the archipelago for any scenario described in this paper. The tool was produced by uploading files generated from the simulation results to the Google Earth Engine platform. The "Atoll Tsunami Explorer" is designed to be user-friendly and targets decision-makers as well as the wider community. To make

interpretation easier, the lagoon boundaries (Spalding et al., 2001) as well as the coastline files used in the simulation (Areas, 2018) are also added to the platform. Simulation results are clipped to the lagoon boundaries for better visualisation, however an option is included that allows visualization of country-wide scenarios.





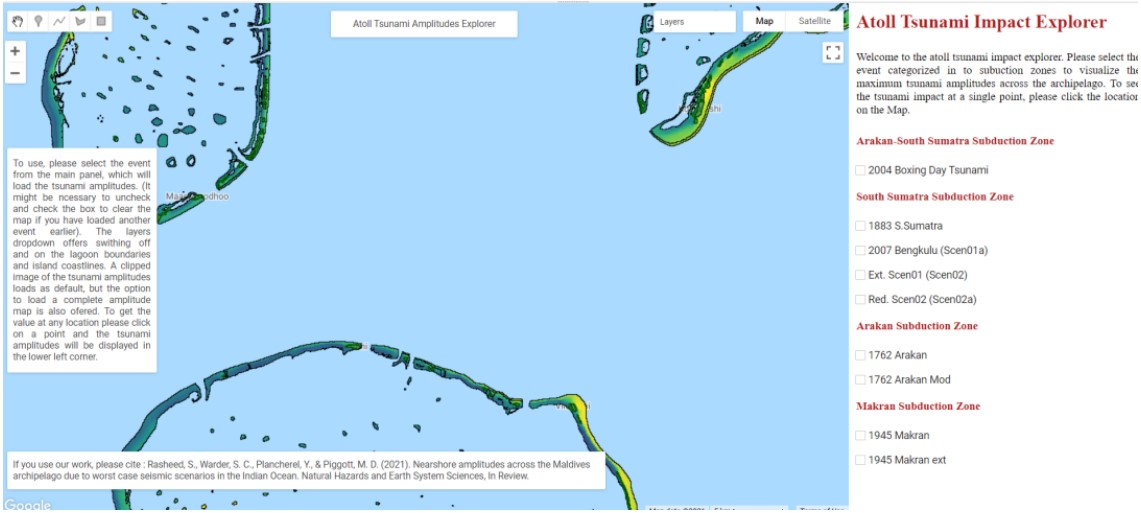

**Figure 9.** Atoll Tsunami Explorer based on the © Google Earth Engine platform, showing the default view with the 2004 tsunami amplitudes visualized with the lagoon and island boundaries.

## 4.3 Limitations of the Study and Recommendations for Improvement

The main limitation of this study is the lack of availability of suitable data both for model setup and validation, including bathymetry and bed friction fields. Even though the bathymetry dataset used (Rasheed et al., 2020) is the highest resolution currently available for the Maldives, it is comprised of data from a variety of sources and the defects of these datasets will manifest as defects in this study. We find that the bathymetry exerts a major influence in directing tsunami waves across the complex geometry of the Atolls of the Maldives, in agreement with Orpin et al. (2016), which showed similar effects in Toekalu in the Pacific. Availability of better bathymetry data, particularly near the coastline, can be used to further improve the model and thus the collection of such data should be a priority for the Maldives as it will have clear and direct benefits for risk management across the archipelago.

In addition to bathymetry data, the model friction parameter also has an influence on predictions of tsunami behaviour. Dilmen et al. (2018) studied the impact of different Manning coefficients in tsunami modelling within a reef environment at Tutuila island in American Samoa, and found that use of a single Manning's $n$ value, as used in this study, was not sufficient to fully account for tsunami-induced flow within a reef environment. Further, Gelfenbaum et al. (2011), predict that beyond a critical reef width, the tsunami amplitude is dominated by the friction parameter, which generally reduces the tsunami amplitude. The availability of data to better describe the friction parameters is needed to facilitate model calibration.

Additionally, the model was run in a UTM (Universal Transverse Mercator) coordinate system, which introduces geometrical errors. However, comparison of the simulation results carried out in section 3.1.2, shows good agreement with field observations. This justifies its use here, especially in light of the other error sources listed above.

Lastly, parameters used in fault models have also been suggested to contribute to overall limitations in tsunami modelling (Poisson et al., 2011). We note that the numerical model used within this work, Thetis, has an adjoint model available. Similar studies for other natural hazards have utilised adjoint methods to perform more complete sensitivity analysis and uncertainty quantification, e.g. Warder et al. (2021). Such methods could be utilised for tsunami hazard assessment to quantify model
uncertainties arising from a variety of sources.

## 5  Conclusions

Detailed high-fidelity tsunami modelling of the Maldives archipelago has been conducted for the first time using a high-resolution bathymetry dataset, with a focus on nearshore maximum tsunami amplitudes. Modelled results, validated against field data obtained for the 2004 tsunami event, show that the model compares well with observational data. Model results
emphasize that tsunami risk assessments for the archipelago need to incorporate high-resolution modelling because Atoll shapes and island geography induce a high level of control over, and thus variability in the resulting tsunami amplitudes due to the development of complex refraction and reflection wave patterns.

On account of the large-scale coastal modifications seen across the Maldives archipelago in recent years (DNP, 2019), future work will need to take into account the possible impact of such large-scale modifications on the natural resilience of the islands
to minimize the impact of far-field tsunamis, as opposed to potentially exacerbating any localised risk. Coastal modifications have already been shown to affect bed stress (Rasheed et al., 2021) across the larger atoll domain as well as at the near shore regions of islands, and the results of this study could be incorporated in such future studies to better understand the resilience of islands to such events.

*Code availability.*

The source code of *Thetis* coastal ocean model used in this study is available from https://thetisproject.org/ as well as https://github.com/thetisproject/thetis.

*Data availability.*

The data presented in this study can be viewed using the Atoll Tsuanmi Explorer App, available from https://zubba1989.users.earthengine as well as within the I-Hex (https://i-hex.github.io/) platform.

*Author contributions.*  S



R ran the simulations, carried out the analysis of results and initiated the writing of the paper. SCW provided support for model setup and analysis of the results. MDP and YP provided supervision, guidance and insights at every stage of the project. All authors participated in the writing and editing of the paper.

*Competing interests.*

The authors declare that they have no conflict of interest.

*Acknowledgements.* The authors would like to acknowledge funding from a Research England GCRF award made to Imperial College London, and the Imperial College London Strategic Priorities Fund 2020-21 awarded for the Island Health Explorer project. SR would like to acknowledge PhD funding from the Islamic Development Bank and Imperial College London.



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
