# Peer review of "Nearshore Tsunami amplitudes across the Maldives archipelago due to worst case seismic scenarios in the Indian Ocean"

_Natural Hazards and Earth System Sciences, 2022_

## Author Comment (AC1)

**Nearshore Tsunami amplitudes across the Maldives archipelago due to worst case seismic scenarios in the Indian Ocean**
**Response to Reviewer One**

Shuaib Rasheed [1], Simon C. Warder [1], Yves Plancherel [1], and Matthew D. Piggott [1]

[1]Department of Earth Science and Engineering, Imperial College London, UK

**Correspondence:** Shuaib Rasheed (s.rasheed18@imperial.ac.uk)

**1 Introduction**

The authors would to like to thank referee 2, for taking the time to review paper providing valuable feedback for improving the manuscript. We are glad that the referee found that providing access to the results via the interactive GEE page, "will be of great benefit to the wider community." Below we respond to each of the comments from the referee and provide a summary of the proposed corrections.

**2 Response to comments**

**2.1 Response to Main Comments**

**2.1.1 Comment Number 01**

Aside from the detailed feedback below my main critique of the work is related to the numerical modelling and would recommend some additional efforts in this regard. The main issue is the use of coarse (50m) fine-resolution' grids. The authors themselves repeatedly state the necessity of high resolution bathymetry/meshes to capture the complex wave patterns of tsunami waves around and within Atolls. They state that their high resolution mesh has a minimum mesh element size of 50m however in their referenced work [Rasheed, 2021 (a)] it appears that bathymetry data on a 10m resolution is available. If high resolution information is key to capturing the complex tsunami wave patterns, something which this reviewer agrees with, why have the authors not used a finer resolution mesh? Is there an issue with computational resources? Please expand on this.

1. *We appreciate that increasing the resolution of the mesh from 50m at the coastline as in the present study to approximately 10m might have implications on the results of the study. We chose 50m due to sensitivity studies done earlier in previous studies where we found that the representation of bathymetry at 50m and 10m did not have considerable difference albeit the increase in mesh elements and computational expense, and also because we are more interested in the wider atoll scale patterns rather than at the island scale. However, as the referee has suggested we understand that it is essential to*

*include a sensitivity study of selecting the mesh resolution to understand the implications on the results of the study and we will include a section in the revised manuscript based on further simulations.*

**2.2 Comment Number 02**

The authors state that the model Thetis can capture wetting/drying using the algorithm described in Eq. 3, however they have chosen a minimum water depth of 0.1m. From this reviewer's experience this minimum depth is overly conservative. If a higher resolution mesh is used than I would encourage the authors to reduce this value. Otherwise the over-topping of low-lying islands may not be captured accurately and thus the influence on the resultant wave pattern will be missed. Further comparisons to run-up and inundation measurements from the 2004 survey could also be made. It should be noted that despite the recognised absence of additional terms in the non-linear shallow water equations (NSWE) for capturing inundation, numerous NSWE solvers have been validated against inundation and runup tasks, [Macias 2017] is one such example.

1. *The authors completely agree with the referee in that avoiding a minimum depth and using wetting and drying to capture inundation would make the study comparable to field measurements. However, during the study we made a pragmatic choice to represent the islands as voids in the mesh, mainly due to the fact that no large scale topographical data is available in the public domain for any of the islands in the Maldives. Some present studies assume fixed heights of 1.5m to 2m for the islands, however from tsunami observational data as reported in the manuscript we find that despite the relative low lying nature of the islands, the relative differences in topographic profiles of the island play a major role in inundation levels across the island. Hence, we decided that due to the lack of data, we would focus on identifying high impact regions at the atoll scale which could be highlighted for further study which could be used to model individual island scale inundations with the availability of high resolution topography and bathymetry. In selecting 0.1m as the minimum depth we also considered the fact that, all most all of the inhabited and industrial islands of the Maldives now have sea walls and additional off shore coastal protection which also needs to be taken to account for island scale modelling.*

**2.3 Comment Number 03**

Has the Thetis model in this set up been validated against traditional tsunami benchmark problems? If not I would suggest taking a look at the problems outlined in [https://nctr.pmel.noaa.gov/benchmark/].

1. *The Thetis coastal ocean model has been benched marked and compared to similar models such as in Pan (2020), where results are presented for some of the standard bench mark cases also outlined in [https://nctr.pmel.noaa.gov/benchmark/]. However, it should be noted that here we used the 2d hydrostatic version of Thetis since we donot focus on inundation and rather on identifying the larger atoll scale patterns of tsunami flow. We appreciate that for more small scale simulations which includes topography and inundation, we would require the use of non-hydrostatic version of Thetis which has been benchmarked in Pan (2020) marked against standard tsunami inundation cases such as the Okushiri tsunami.*

**2.4 Response to Minor Comments**

1. *Since these are suggestions by the referee to make minor changes to the manuscript to fix figures and typing errors, we will make the changes as suggested in the revised manuscript.*

**3   Conclusion**

We have proposed to include a sensitivity test to address the main issue of the reviewer in selecting the mesh resolution. Further, we have tried to explain the choice of using a minimum depth and excluding inundation modelling for this study due to lack of data. We hope that these proposed changes are satisfactory to the referee.

**References**

60 Pan, W.: Development of a non-hydrostatic coastal ocean model using the discontinuous Galerkin method, 2020.

---

## Author Comment (AC2)

**Nearshore Tsunami amplitudes across the Maldives archipelago due to worst case seismic scenarios in the Indian Ocean Response to Reviewer Two**

Shuaib Rasheed [1], Simon C. Warder [1], Yves Plancherel [1], and Matthew D. Piggott [1]

[1]Department of Earth Science and Engineering, Imperial College London, UK

**Correspondence:** Shuaib Rasheed (s.rasheed18@imperial.ac.uk)

**1  Introduction**

The authors would like to thank the reviewer for taking the time to provide a comprehensive review of the submitted manuscript. We are glad that the reviewer found that "The work is significant, the presentation clear and well structured and the methods mostly up to date." and that "with relatively little effort, this work could in my view be of even more relevance."

5    Below, we respond to each of the comments from the reviewer, and provide a summary of the corresponding changes proposed to be made.

**2  Response to comments**

**2.1  Response to Major Comments**

**2.1.1  Comment Number 01**

10  A short discusion on why not a standard tsunami model was used, but a self built non-validated (at least not with the standard tsunami benchmarks according to Synolakis et al., 2008) based on Firedrake. What are the advantages compared to e.g. COMCOT or TsunaCLAW?

1. *The Thetis coastal ocean model has been benched marked and compared to similar models such as in Pan (2020), where results are presented for some of the standard bench mark cases also outlined in [https://nctr.pmel.noaa.gov/benchmark/].*
15    *However, it should be noted that here we used the 2d hydrostatic version of Thetis since we donot focus on inundation and rather on identifying the larger atoll scale patterns of tsunami flow. We appreciate that for more small scale simulations which includes topography and inundation, we would require the use of non-hydrostatic version of Thetis which has been benchmarked in Pan (2020) marked against standard tsunami inundation cases such as the Okushiri tsunami. In line with the proposed comment by the reviewer we will add a section to highlight the choice.*

**2.2 Comment Number 02**

2. What kind of criteria were used for the diverse decisions made: a. Mesh refinement - is it just proximity to coast? b. removal of islands from the large scale simulation - is it size?

1. *In selecting the mesh resolutions we made use of a sensitivity study carried out earlier, where we found that a mesh resolution of 100m was adequate to capture the bathymetry of the shallow lagoons (Rasheed et al., 2021). However, based on the reviewer feedback we propose to carryout a sensitivity study to understand the impact of varying mesh resolutions on the results.*

2. *The islands were removed from the simulations because inundation modelling was not considered in the simulations. This was mainly due to the fact that topographic data for the islands of the Maldives is not available in the public domain. If we were to model the islands as flat, as in some studies it would not match with field observations, where we find that the relative differences in island topographies produces differences in tsunami inundation heights across the islands. Hence, a pragmatic choice was made to remove the islands and focus on identifying regions where tsunami amplitudes were high which then could be marked as areas for future studies.*

**2.3 Comment Number 03**

To me the local resolution mesh sizes seem still rather large. A 5000 m mesh size at the Maledives Atoll coast for the large-scale simulation yields an effective wave lenght representation of 30 km or more (given the linear P1 elements of the DG discretization). Is this a reasonable scale? Additionally, the non-uniform mesh would allow for higher local resolution without much additional effort in terms of added unknowns, since the local area of refinement would cover only fractions of the domain. The same applies to the local simulations, where a 50 m mesh size allows to represent wave lengths of approx. 500 m or a little less that that. With island sizes of only meters in size, I doubt if this is high enough a resolution for quantitatively accurate results. Some sensitivity studies would be helpful in this.

1. *As the reviewer has suggested, we agree that these resolutions are large. However, we would like to highlight that these are mesh resolutions of the larger Indian ocean scale simulations carried out. Also, we would like to add that the larger simulation was carried out using GEBCO bathymetry which does not include any complex features of the Maldives archipelago and as thus increasing the mesh resolution with GEBCO bathymetry would not provide any advantages despite the additional computational time. For the nested simulation which covered the Maldives archipelago and utilized a higher resolution dataset which have the complex features present, we used 100m resolution at the lagoon scale. Since the atoll peripheries are also lagoons, this ensures that the atoll boundaries are also in 100m resolution. However, as the reviewer has suggested sensitivity studies will be helpful to highlight these choices we will include a section on the proposed revision to the manuscript.*

**2.4 Comment Number 04**

Since you indicated in the text that you are only considering wave heights at the coast and no inundation, what are the boundary conditions at the coasts then? In Harig et al. (2008) it was found that inundation BC are necessary even if not used to realistically prepresent coastal reflection of waves.

1. *Here, since we donot take inundation to account we set the coastline boundaries to no normal flow velocity. We have not taken to account the implication of the coastline boundary condition since the island sizes are relatively very small in comparison to even the atoll scale, and we find that the the impact of shallow lagoons on the periphery and the inner basin of the atolls have a much more significant impact on the flow patterns. However, we will carry out further simulations to check if changing the boundary condition at the coastline has any impact on the flow patterns.*

**2.5 Comment Number 05**

In order to evaluate the wave build-up it would also be valuable to consider the different wave lengths/periods in comparison to the obstacle size (atoll diameter e.g.) to have a conceptual understanding of this phenomenon. I hypothesize that a singular atoll of a size less than - say - half the deep ocean wave length will be passed by the wave without major harm, given the extremely steep bathymetry. But this would be an interesting topic of diagnostics, analysis and discussion for the different locations and angles of attack.

1. *The authors agree that this is a very interesting question, and indeed as we have demonstrated in the paper along with the field observations, due to the steepness of the atolls, tsunami waves passed across the Maldives with relatively low amplitudes. We would like to highlight that as we have mentioned across the manuscript the tsunami build up across some regions of the county occurred mainly due to the geometric shape of the atolls it self and the location, which forced tsunami to propagate at high velocity at these regions.*

**2.6 Comment Number 06**

You claim that such results are only possible by high resolution bathymetry data and go further to ask for even higher resolution in this respect. But you do not prove that this is really the case. It would be very instructive (and in your case probably easily possible) to actually demonstrate this claim by comparing the effect of diffraction, reflection and deflection in your large-scale and small-scale simulations. For example the results in figure 8, do they differ substantially for your large- and small-scale simulations? If so, I would buy your demand for ever higher resolution ;-) Here I assume that you use the same bathymetry data in your simulations, but that you interpolate to your unstructured mesh and therefore have different discrete bathymetries in your simulations.

1. *We agree with the reviewer that it was an oversight to not have compared the larger scale tsunami simulation carried out with GEBCO bathymetry with the smaller Maldives scale simulation carried out with a much more high resolution bathymetry. We have addressed this issue in past papers such as in Rasheed et al. (2020), where we showed that low*

*resolution bathymetry is not adequate to capture the flow patterns across the complex bathymetry of the atolls of the Maldives because these datasets are completely devoid of features which actually give rise to these flow patterns. As suggested by the reviewer we will add this comparison in the revised manuscript.*

**2.7 Response to Minor Comments**

85      1. *Since these are suggestions by the reviewer to make minor changes to the manuscript to fix figures and typing errors, we will make the changes as suggested in the revised manuscript.*

**3   Conclusion**

The main issue that the review has raised has to do with the choice of mesh resolutions selected for the simulations. We have attempted to answer the queries of the reviewer with the main justification that we have two simulations of two different scales
90 with two different bathymetry datasets. However, as the reviewer has suggested, it would be helpful to include a sensitivity study we propose that we will add a section with additional simulations corresponding to a sensitivity study. We hope that these responses are to the satisfaction of the reviewver.

**References**

Pan, W.: Development of a non-hydrostatic coastal ocean model using the discontinuous Galerkin method, 2020.

95    Rasheed, S., Warder, S. C., Plancherel, Y., and Piggott, M. D.: An Improved Gridded Bathymetric Dataset and Tidal Model for the Maldives Archipelago, Earth and Space Science, p. e2020EA001207, 2020.

Rasheed, S., Warder, S. C., Plancherel, Y., and Piggott, M. D.: Response of tidal flow regime and sediment transport in North Malé Atoll, Maldives, to coastal modification and sea level rise, Ocean Science, 17, 319–334, 2021.

---

## Referee Report (RR1)

Nat. Hazards Earth Syst. Sci. Discuss., referee comment RC1
https://doi.org/10.5194/nhess-2022-95-RC1, 2022

**Comment on nhess-2022-95**

Anonymous Referee #1

Referee comment on "Nearshore Tsunami amplitudes across the Maldives archipelago due to worst case seismic scenarios in the Indian Ocean" by Shuaib Rasheed et al., Nat. Hazards Earth Syst. Sci. Discuss., https://doi.org/10.5194/nhess-2022-95-RC1, 2022

This reviewer would like to commend the authors on their work. Efforts associated with quantifying tsunami hazards/vulnerability is a welcome addition and this work represents first of its kind tsunami simulations for the Maldives archipelago. Further, I would particularly like to commend their efforts on providing an online server to access the results, as stated by the authors this will be of great benefit to the wider community.

However, before I can recommend this paper for publication I would encourage the authors to engage with the feedback and comments provided below.

Main Comments

- Aside from the detailed feedback below my main critique of the work is related to the numerical modelling and would recommend some additional efforts in this regard. The main issue is the use of coarse (50m) `fine-resolution' grids. The authors themselves repeatedly state the necessity of high resolution bathymetry/meshes to capture the complex wave patterns of tsunami waves around and within Atolls. They state that their high resolution mesh has a minimum mesh element size of 50m however in their referenced work [Rasheed, 2021 (a)] it appears that bathymetry data on a ~10m resolution is available. If high resolution information is key to capturing the complex tsunami wave patterns, something which this reviewer agrees with, why have the authors not used a finer resolution mesh? Is there an issue with computational resources? Please expand on this.
- The authors state that the model Thetis can capture wetting/drying using the algorithm described in Eq. 3, however they have chosen a minimum water depth of 0.1m. From this reviewer's experience this minimum depth is overly conservative. If a higher resolution mesh is used than I would encourage the authors to reduce this value. Otherwise the overtopping of low-lying islands may not be captured accurately and thus the influence on the resultant wave pattern will be missed. Further comparisons to runup and inundation measurements from the 2004 survey could also be made. It should be noted that despite the recognised absence of additional terms in the non-linear shallow water equations (NSWE) for capturing inundation, numerous NSWE solvers have been validated against inundation and runup tasks, [Macias 2017] is one such example.

Line by Line Comments

Section 1.

Line 25: Rephrase ``These data imply …"

Line 30 - 35: It might be worth mentioning the return periods of some of the findings

Line 38: Please expand on the ``safe island concept" for those who are unaware of it.

Line 41-42: Rephrase ``impacted less and others being impacted more"

Lines 50-55: Please make reference of [Xie 2019], where a tsunami hazard assessment of the Xisha Archipelago was carried out.

Section 2.

Line 102: Typo ``Boxing Day even" should be ``Boxing Day event"

Line 113: Typo ``Table 1" should be ``Table 2"

Line 113: Rephrase ``worst most-likely"

Section 2.2: It would be good to make reference to Figure 1, which showcases the location of the various sources

Section 2.3: It appears that this section is a direct copy of a section in the authors previous works [Rasheed, 2021 (a) and Rasheed, 2021 (b)].

Line 150: Units for kinematic viscosity

Line 152: I would suggest introducing tau and rho here instead of on Line 160. Please also make reference to that fact that the value of n will be discussed in section 2.3.4.

Line 153: Please provide a further explanation for the $P^{DG}_1 - P^{DG}_1$ term.

Comment: Has the Thetis model in this set up been validated against traditional tsunami benchmark problems? If not I would suggest taking a look at the problems outlined in [https://nctr.pmel.noaa.gov/benchmark/].

Line 166: ``Each simulation was run in a full simulation which spatially extended ..'' It is not clear from this sentence how the nested simulations were carried out, please clarify this.

Line 166-167: Reorder the Figure numbers, ``Figures 2 (a) and 1 (a)'' and ``Figures 2 (b) and 1 (b)''.

Line 171-173: The authors state that the tidal variation of 1m is ``very small'' and can therefore be discounted however in later sections they state ``If combined with a high tide, tsunamis generated from scenario 2a would likely have an impact across locations predicted to have higher amplitudes''. I would agree with the later and state that tidal forcing can play a role in inundation levels. Not including tidal forcing in the study is acceptable but please correct this section and make note of the limitation in section 4.3 (Limitations of the Study and Recommendations for Improvement).

Figure 1: Please increase the size of Figure 1 (b). It is difficult to make out the high resolution features. Please also ensure that all x and y axis are labelled.

Figure 1: Typo in the caption ``13 orth Nilandhe'' and ``20. ddu Atoll''.

Figure 2: Please make the subplot (b) larger and label the axis correctly.

Section 2.3.5: How are the full and higher resolution mesh merged? From reading it appears there is a mismatch in mesh resolution at the boundaries, with the full mesh having a resolution of 5km to 7.5km while the nested mesh has a resolution of 10km at the boundaries. Please provide some details on nested procedure.

Section 3.

Figure 3: Please make subplots (a) and (b) larger. There appears to be an artefact of the nesting in the bottom right of both subplots (a) and (b). The bottom right corner exhibits some high wave heights which exhibit a discontinuous drop-off when moving in a north western direction. Is this physical or an artefact of the nesting?

Section 3.1.2: Why have you chosen a Pearson correlation coefficient? Would a RMSE value be more appropriate? Please explain.

Lines 260 -266: This appears to be a one sided comparison. What about the areas of low impact? Do the simulated and observed areas of low impacts also match up? This is an equally interesting comparison.

Figure 4. It might be useful to provide a map showing where these subplots are located in the Maldives. Please provide a wave height scale and label the axis for each subplot.

Figure 6 and 7: Typo ``worst case fault case"

Section 4.

Lines 347-349: Are the refraction, diffraction and reflection patterns repeated across different simulated sources? Please comment on this.

Figure 8: Mark the Atolls (South Nilande, Mulaku and Kolhumadula) in subplot (a). Please re-plot with the wave height coloring centred on 0m. It may be interesting to see reflections etc.

Line 352: Please reference the work of [Reymond, 2012], where the role of reef systems on the amplification of tsunami waves is captured as a site specific amplification parameter.

Lines 353-357: Please rephrase this sentence.

Line 372: Can you please qualify ``with high tsunami flow velocities across the shallow and narrow channels", as there are no plots explicitly showing this behaviour.

Section 4.2: As stated above I highly commend the authors for providing the online explorer. However the links do not work and I was unable to access the server. Please correct this.

Line 450: The following work should be cited as an additional approach for investigating uncertainties [Giles, 2021]. In that work the uncertainty on the source is propagated to maximum wave heights using cheap statistical emulators.

Section 5.

Line 456-457: The statement ``variability in the resulting tsunami amplitudes due to the development of complex refraction and reflection wave patterns" should be qualified with further results such as those shown in section 4.1

To finish I would like to reiterate my commendation of the authors efforts and appreciate that my main comments listed above may be deemed harsh. However, if the high resolution data is available I would encourage the authors to re simulate at a higher resolution.

References

- Rasheed, 2021 (a): Rasheed, S., Warder, C. S., Plancherel, Y. and Piggott, M. D: "An Improve Gridded Bathymetric Data Set and Tidal Model for the Maldives Archipelago", Earth and Space Science, 8, 5, 2021
- Macias, 2017: Macias, J., Castro, M. J., Ortega, S., Escalante, C., Gonzalez-Vida, J. M.: "Performance Benchmarking of Tsunami-HySEA Model for NTHMP's Inundation Mapping Activities", Pure and Applied Geophysics, 8, 3147--3183, 2017
- Xie, 2019: Xie, X., Chen, C.,  Li, L.,  Wu, S., Yuen, D. A., Wang, D.:"Tsunami hazard assessment for atoll islands inside the South China Sea: A case study of the Xisha Archipelago", Physics of the Earth and Planetary Interiors, 290, 2019
- Rasheed, 2021 (b): Rasheed, S., Warder, C. S., Plancherel, Y. and Piggott, M. D: "Response of tidal flow regime and sediment transport in North Male Atoll, Maldives, to coastal modification and sea level rise", Ocean Sci., 17, 319-334, 2021
- Reymond, 2012: Reymond, D. and Okal, E. A. and H{\'{e}}bert, H. and Bourdet, M.: "Rapid forecast of tsunami wave heights from a database of pre-computed simulations, and application during the 2011 Tohoku tsunami in French Polynesia", Geophysical Research Letters, 11, 1—6, 2012
- Giles, 2021: Giles, D. and Gopinathan, D. and Guillas, S. and Dias, F.: "Faster Than Real Time Tsunami Warning with Associated Hazard Uncertainties", Frontiers in Earth Science, 8, 2021

---

## Author Response (AR2)

**Nearshore Tsunami amplitudes across the Maldives archipelago due to worst case seismic scenarios in the Indian Ocean Response to Reviewers**

Shuaib Rasheed [1], Simon C. Warder [1], Yves Plancherel [1], and Matthew D. Piggott [1]

[1]Department of Earth Science and Engineering, Imperial College London, UK

**Correspondence:** Shuaib Rasheed (s.rasheed18@imperial.ac.uk)

**1 Introduction**

We would like to thank both referees for taking the time to re-review the manuscript and provide feedback on how to further improve the manuscript. Below, we respond to each of the comments from the reviewers, and provide a summary of the corresponding changes made.

**2 Response to comments (Reviewer One)**

**2.1 Response to Major Comments**

**2.1.1 Comment Number 01**

One of my major questions of the first review is still not answered: What is the criterion for mesh refinement? I understand that 100 m finest mesh is based on "sensitivity studies", however the question is still what was the criterion to refine? Maybe you could describe it somewhere?

1. *The newly included section 3.2.2 provides details of the mesh sensitivity study that was undertaken to study the impact of the sensitivity of the model to the mesh resolution at the lagoons. Figures 4 and 5 provides details of these studies. Figure 4 shows the comparison of the field observations with the simulated maximum tsunami elevations for different mesh element sizes at the lagoons. With increased mesh refinement at the lagoon an increased correlation is clearly observed. Further, the capability of the mesh to represent the bathymetry was also studied. The results shows that with increasing mesh resolution, an increase in the correlation with the actual bathymetry is observed. The associated text from the revised paper is quoted here for the convenience of the reviewer.*

*Figure 4 presents the results of the numerical sensitivity study carried out using the fault model given by Grilli et al. (2007). The correlation metric is based on comparing the simulated and observed maximum amplitudes from across the archipelago, as discussed in Section 3.2. The results show that model outputs are very sensitive to the mesh resolutions used at the lagoons. The use of 100m mesh resolution at the lagoon boundaries produces a correlation of approximately 0.9, while using larger mesh element sizes at the lagoon produces significantly lower correlations. Based on these results, we proceed with the mesh featuring 100m resolution at the lagoons. While the sensitivity study suggests that, refinement of the mesh at the lagoon boundaries even further will improve the correlation, as seen in table **??**, this will result in a very large number of mesh elements, making the computational cost prohibitive. Further, as the island coastlines within the lagoons are meshed with a resolution of 50 m, the presence of islands within the lagoons also contribute to the improvement of overall mesh sizes. Initially, we test the capacity of the meshes to accurately represent the bathymetry of the domain. This is of particular interest here as the complex bathymetry of the domain is predicted to govern the tsunami flow pattern across the domain. We linearly interpolated the high resolution bathymetry on to each of the meshes. The correlation between the meshes and the high resolution original bathymetry given in Figure 5, shows that all of the meshes used for the simulation were able to represent the bathymetry with a high degree of correlation, with the usage of 100m mesh lengths across the lagoons providing up to 90% correlation, which is in line with results from Rasheed et al. (2021) for a single administrative atoll in comparison to the entire archipelago considered here. [Section 3.2.2]*

**2.2 Comment Number 02**

20  Why do you still describe the treatment of wetting and drying (page 7, line 158 ff.), but then in the later text (e.g. page 12, line 260) you exclude inundation results. This is very confusing. Does the model include inundation or not? Maybe it is just a wording issue? Your calculations use inundation, but the inundation results on the islands are not quantitatively useful, since the topography data is not accurate enough? Then please write it so.

1. *As highlighted by the reviewer we describe the wetting and drying treatment in the manuscript because even though the*
25  *model does not consider inundation, i.e. we do not mesh above mean sea level onto islands due to a lack of data, we find that allowing for the wetting and drying of initially wet areas below mean sea level increases model stability and means that we do not need to artificially deepen shallow regions to ensure the meshed region is always wet. Furthermore, we have now specifically stated in the manuscript that inundation is not considered in the study mainly because the topography data is not available and that the results from this study could be used to identify and carryout detailed*
30  *tsunami assessments of specific islands of interest as part of a future study. These are now included in different parts of the manuscript as follows :*

*Although inundation is not the focus of this work, we found that the introduction of a small minimum depth decreases the computation time of the simulation. [Section 2.3]*

*Survey results from all of these field studies are included here. However, where the spatial distances between the field obser-vations are very small, we considered the measurement which was closest to the coastline, since the model used here does not capture inundation of islands as explicit simulations of inland flooding would require high-resolution data capturing land features, which is not currently available. [Section 3.2]*

*Additionally, topographic data, currently unavailable would also provide a means of incorporating inundation modelling, not considered in this study. [Section 4.3]*

**2.3 Comment Number 03**

Mesh size sensitivity analysis is very welcome. However, your experiments show that even higher resolutions would be prefer-able, since the correlation with 100 m local mesh size is not yet saturated. Maybe you could either run one more higher resolution test (even locally) or you at least discuss this.

1. *We appreciate that the resolution at the lagoon boundaries could be reduced further. However, as seen in Table 3, increasing the mesh resolution at the lagoons increases the number of mesh elements and nodes, and given the very large spatial boundary and the number of lagoons, beyond a resolution of 100m at the lagoons it is computationally not feasible to do so with the current computational power available. Further, it should also be noted that even though 100 m is the mesh resolution at the lagoon boundary, the mesh resolution at the coastline of the islands within the lagoons are at 50 m, which further improves the overall mesh resolution across the lagoon. We add the following acknowledgement of this fact and note further that a detailed single island study at higher resolutions is the subject of separate work.*

*Based on these results, we proceed with the mesh featuring 100m resolution at the lagoons. While the sensitivity study suggests that, refinement of the mesh at the lagoon boundaries even further may improve the correlation, as seen in table 3, this will result in a very large number of mesh elements, making the computational cost prohibitive. Further, as the island coastlines within the lagoons are meshed with a resolution of 50 m, the presence of islands within the lagoons also contribute to the improvement of overall mesh sizes. [Section 3.2]*

**2.4 Response to Minor Comments**

1. *Since these are suggestions by the reviewer to make minor changes to the manuscript to fix figures and typing errors, we will make the changes as suggested in the revised manuscript.*

**3 Conclusion**

The main issue that the reviewers has raised has to do with the choice of mesh resolutions selected for the simulations. We have made changes to the manuscript to further explain the reasonsing behind the choices for the mesh resolutions, and attempted to answer the queries of the reviewers and hope that these are to the satisfaction of the reviewer.

**4 Response to comments (Reviewer Two)**

As the review two has flagged the same issues as earlier, here we provide additional details on how the manuscript has been updated to address the feedback provided by the reviewer.

**4.1 Response to Major comments**

**4.1.1 Comment Number 01**

A short discussion on why not a standard tsunami model was used, but a self built non-validated (at least not with the standard tsunami benchmarks according to Synolakis et al., 2008) based on Firedrake. What are the advantages compared to e.g. COMCOT or TsunaCLAW?

1. *The Thetis coastal ocean model has been benchmarked and compared to similar models such as in Pan (2020), where results are presented for some of the standard benchmark cases also outlined in [https://nctr.pmel.noaa.gov/benchmark/]. However, it should be noted that here we used the 2d hydrostatic version of Thetis since we do not focus on inundation and rather on identifying the larger atoll scale patterns of tsunami flow. We appreciate that for more small scale simulations which include very complex topography and inundation processes, we would require the use of the non-hydrostatic version of Thetis which has been benchmarked in Pan (2020) marked against standard tsunami inundation cases such as the Okushiri tsunami. In line with the proposed comment by the reviewer we have added the following in the manuscript to highlight this as a future work:*

> *The availability of topographic data will also enable higher resolution nearshore simulations using the 3D, non-hydrostatic version of Thetis Pan et al. (2019), which has been validated against standard tsunami benchmarks, at geographical regions of interest identified from this large scale study. [Section 4.3]*

**4.2 Comment Number 02**

2. What kind of criteria were used for the diverse decisions made: a. Mesh refinement - is it just proximity to coast? b. removal of islands from the large scale simulation - is it size?

1. *The choice of mesh refinement and selection has now been explained in detail via additional simulations carried out as part of a sensitivity study, further explained in section 3.2.2 as follows.*

> *Figure 4 presents the results of the numerical sensitivity study carried out using the fault model given by Grilli et al. (2007). The correlation metric is based on comparing the simulated and observed maximum amplitudes from across the archipelago, as discussed in Section 3.2. The results show that model outputs are very sensitive to the mesh resolutions used at the lagoons. The use of 100m mesh resolution at the lagoon boundaries produces a correlation of approximately 0.9, while using larger mesh element sizes at the lagoon produces significantly lower correlations. Based on these results, we proceed with the mesh featuring 100m resolution at the lagoons. While the sensitivity study suggests that, refinement of the mesh at the lagoon boundaries even further will improve the correlation, as seen in table ??, this will result in a very large number of mesh elements, making the computational cost prohibitive. Further, as the island coastlines within the lagoons are meshed with a resolution of 50 m, the presence of islands within the lagoons also contribute to the improvement of overall mesh sizes. Initially, we test the capacity of the meshes to accurately represent the bathymetry of the domain. This is of particular interest here as the complex bathymetry of the domain is predicted to govern the tsunami flow pattern across the domain. We linearly interpolated the high resolution bathymetry on to each of the meshes. The correlation between the meshes and the high resolution original bathymetry given in Figure 5, shows that all of the meshes used for the simulation were able to represent the bathymetry with a high degree of correlation, with the usage of 100m mesh lengths across the lagoons providing up to 90% correlation, which is in line with results from Rasheed et al. (2021) for a single administrative atoll in comparison to the entire archipelago considered here. [Section 3.2.2]*

2. *The islands were removed from the simulations because inundation modelling was not considered in the simulations. This was mainly due to the fact that topographic data for the islands of the Maldives is not available in the public domain. If we were to model the islands as flat, as in some studies it would not match with field observations, where we find that the relative differences in island topographies produces differences in tsunami inundation heights across the islands. Hence, a pragmatic choice was made to remove the islands and focus on identifying regions where tsunami amplitudes were high which then could be marked as areas for future studies. As highlighted earlier we have now explicitly stated this within the manuscript.*

3. > *Additionally, topographic data, currently unavailable would also provide means of incorporating inundation modelling, not considered in this study.. [Section 4.3]*

**4.3 Comment Number 03**

To me the local resolution mesh sizes seem still rather large. A 5000 m mesh size at the Maldives Atoll coast for the large-scale simulation yields an effective wave length representation of 30 km or more (given the linear P1 elements of the DG discretization). Is this a reasonable scale? Additionally, the non-uniform mesh would allow for higher local resolution without much additional effort in terms of added unknowns, since the local area of refinement would cover only fractions of the domain. The same applies to the local simulations, where a 50 m mesh size allows to represent wave lengths of approx. 500 m or a little less that that. With island sizes of only meters in size, I doubt if this is high enough a resolution for quantitatively accurate results. Some sensitivity studies would be helpful in this.

1. *As the reviewer has suggested we have now included a sensitivity study to highlight the choices of mesh resolution selection as described earlier in this response document. Further, as highlighted this study provides the basis for more in depth local island scale modelling at areas of interest.*

**4.4 Comment Number 04**

Since you indicated in the text that you are only considering wave heights at the coast and no inundation, what are the boundary conditions at the coasts then? In Harig et al. (2008) it was found that inundation BC are necessary even if not used to realistically represent coastal reflection of waves.

1. *Here, since we do not take inundation into account we set the coastline boundaries to no normal flow velocity. We have not taken into account the implication of the coastline boundary condition since the island sizes are relatively very small in comparison to even the atoll scale, and we find that the impact of shallow lagoons on the periphery and the inner basin of the atolls have a much more significant impact on the flow patterns. (For comparison the islands occupy less than one percent of the domain compared to the ocean).*

**4.5 Comment Number 05**

In order to evaluate the wave build-up it would also be valuable to consider the different wave lengths/periods in comparison to the obstacle size (atoll diameter e.g.) to have a conceptual understanding of this phenomenon. I hypothesize that a singular atoll of a size less than - say - half the deep ocean wave length will be passed by the wave without major harm, given the extremely steep bathymetry. But this would be an interesting topic of diagnostics, analysis and discussion for the different locations and angles of attack.

1. *The authors agree that this is a very interesting question, and indeed as we have demonstrated in the paper along with the field observations, due to the steepness of the atolls, tsunami waves passed across the Maldives with relatively low amplitudes. We would like to highlight that as we have mentioned across the manuscript the tsunami build up across some regions of the county occurred mainly due to the geometric shape of the atolls themselves and the location, which forced tsunami to propagate at high velocity at these regions. We also agree that it would indeed be an interesting topic to further study the relative impact of different angles of tsunami attack on the islands as part of a future study. We should highlight that this study provides an insight on how the angle of attack of the tsunami is different for different islands, depending on the location as discussed for the islands of Gemendhoo, Rinbudhoo etc, with tsunami field data corroborating with the simulation results.*

**4.6 Comment Number 06**

You claim that such results are only possible by high resolution bathymetry data and go further to ask for even higher resolution in this respect. But you do not prove that this is really the case. It would be very instructive (and in your case probably easily possible) to actually demonstrate this claim by comparing the effect of diffraction, reflection and deflection in your large-scale

and small-scale simulations. For example the results in figure 8, do they differ substantially for your large- and small-scale simulations? If so, I would buy your demand for ever higher resolution ;-) Here I assume that you use the same bathymetry data in your simulations, but that you interpolate to your unstructured mesh and therefore have different discrete bathymetries in your simulations.

1. *We agree with the reviewer that it was an oversight to not have compared the larger scale tsunami simulation carried out with GEBCO bathymetry with the smaller Maldives scale simulation carried out with a much more high resolution bathymetry. We have partially addressed this issue in past papers such as in Rasheed et al. (2020), where we showed that low resolution bathymetry is not adequate to capture the flow patterns across the complex bathymetry of the atolls of the Maldives because these datasets are completely devoid of features which actually give rise to these flow patterns. This has been further detailed within the manuscript in section 2.3.2*

**4.7 Response to Minor Comments**

1. *Since these are suggestions by the reviewer to make minor changes to the manuscript to fix figures and typing errors, we will make the changes as suggested in the revised manuscript.*

**References**

Pan, W.: Development of a non-hydrostatic coastal ocean model using the discontinuous Galerkin method, 2020.

Pan, W., Kramer, S. C., and Piggott, M. D.: Multi-layer non-hydrostatic free surface modelling using the discontinuous Galerkin method, Ocean Modelling, 134, 68 – 83, https://doi.org/https://doi.org/10.1016/j.ocemod.2019.01.003, http://www.sciencedirect.com/science/article/pii/S1463500318302312, 2019.

Rasheed, S., Warder, S. C., Plancherel, Y., and Piggott, M. D.: An Improved Gridded Bathymetric Dataset and Tidal Model for the Maldives Archipelago, Earth and Space Science, p. e2020EA001207, 2020.

140

---

## Author Response (AR3)

**NEAR SHORE IMPACT OF FAR FIELD TSUNAMIS ACROSS THE MALDIVES ARCHIPELAGO Response to Reviewer**

Shuaib Rasheed [1], Simon C. Warder [1], Yves Plancherel [1], and Matthew D. Piggott [1]

[1]Department of Earth Science and Engineering, Imperial College London, UK

**Correspondence:** Shuaib Rasheed (s.rasheed18@imperial.ac.uk)

**1   Introduction**

The authors would to like to thank referee, for taking the time to review paper and providing valuable feedback for improving the manuscript. We also apologise for the inconvenience in having mixed up the reviewers. Here we provide feedback on the outstanding comments from round one that the reviewer has requested to be further addressed.

**2   Response to comments**

**2.1   Response to Main Comments**

**2.1.1   Comment Number 01**

Aside from the detailed feedback below my main critique of the work is related to the numerical modelling and would recommend some additional efforts in this regard. The main issue is the use of coarse (50m) fine-resolution' grids. The authors themselves repeatedly state the necessity of high resolution bathymetry/meshes to capture the complex wave patterns of tsunami waves around and within Atolls. They state that their high resolution mesh has a minimum mesh element size of 50m however in their referenced work [Rasheed, 2021 (a)] it appears that bathymetry data on a  10m resolution is available. If high resolution information is key to capturing the complex tsunami wave patterns, something which this reviewer agrees with, why have the authors not used a finer resolution mesh? Is there an issue with computational resources? Please expand on this.

1. *The newly included section 3.2.2 provides details of the mesh sensitivity study that was undertaken to study the impact of the sensitivity of the model to the mesh resolution at the lagoons. Figures 4 and 5 provides details of these studies. Figure 4 shows the comparison of the field observations with the simulated maximum tsunami elevations for different mesh element sizes at the lagoons. With increased mesh refinement at the lagoon an increased correlation is clearly observed. Further, the capability of the mesh to represent the bathymetry was also studied. The results shows that with increasing mesh resolution, an increase in the correlation with the actual bathymetry is observed. The associated text from the revised paper is quoted here for the convenience of the reviewer.*

*Figure 4 presents the results of the numerical sensitivity study carried out using the fault model given by Grilli et al. (2007). The correlation metric is based on comparing the simulated and observed maximum amplitudes from across the archipelago, as discussed in Section 3.2. The results show that model outputs are very sensitive to the mesh resolutions used at the lagoons. The use of 100m mesh resolution at the lagoon boundaries produces a correlation of approximately 0.9, while using larger mesh element sizes at the lagoon produces significantly lower correlations. Based on these results, we proceed with the mesh featuring 100m resolution at the lagoons. While the sensitivity study suggests that, refinement of the mesh at the lagoon boundaries even further will improve the correlation, as seen in table ??, this will result in a very large number of mesh elements, making the computational cost prohibitive. Further, as the island coastlines within the lagoons are meshed with a resolution of 50 m, the presence of islands within the lagoons also contribute to the improvement of overall mesh sizes. Initially, we test the capacity of the meshes to accurately represent the bathymetry of the domain. This is of particular interest here as the complex bathymetry of the domain is predicted to govern the tsunami flow pattern across the domain. We linearly interpolated the high resolution bathymetry on to each of the meshes. The correlation between the meshes and the high resolution original bathymetry given in Figure 5, shows that all of the meshes used for the simulation were able to represent the bathymetry with a high degree of correlation, with the usage of 100m mesh lengths across the lagoons providing up to 90% correlation, which is in line with results from Rasheed et al. (2021) for a single administrative atoll in comparison to the entire archipelago considered here. [Section 3.2.2]*

2. *We appreciate that the resolution at the lagoon boundaries could be reduced further. However, as seen in Table 3, increasing the mesh resolution at the lagoons increases the number of mesh elements and nodes, and given the very large spatial boundary and the number of lagoons, beyond a resolution of 100m at the lagoons it is computationally not feasible to do so with the current computational power available. Further, it should also be noted that even though 100 m is the mesh resolution at the lagoon boundary, the mesh resolution at the coastline of the islands within the lagoons are at 50 m, which further improves the overall mesh resolution across the lagoon. We add the following acknowledgement of this fact and note further that a detailed single island study at higher resolutions is the subject of separate work.*

*Based on these results, we proceed with the mesh featuring 100m resolution at the lagoons. While the sensitivity study suggests that, refinement of the mesh at the lagoon boundaries even further may improve the correlation, as seen in table 3, this will result in a very large number of mesh elements, making the computational cost prohibitive. Further, as the island coastlines within the lagoons are meshed with a resolution of 50 m, the presence of islands within the lagoons also contribute to the improvement of overall mesh sizes. [Section 3.2]*

**2.2 Comment Number 02**

The authors state that the model Thetis can capture wetting/drying using the algorithm described in Eq. 3, however they have chosen a minimum water depth of 0.1m. From this reviewer's experience this minimum depth is overly conservative. If a higher resolution mesh is used than I would encourage the authors to reduce this value. Otherwise the over-topping

35     of low-lying islands may not be captured accurately and thus the influence on the resultant wave pattern will be missed. Further comparisons to run-up and inundation measurements from the 2004 survey could also be made. It should be noted that despite the recognised absence of additional terms in the non-linear shallow water equations (NSWE) for capturing inundation, numerous NSWE solvers have been validated against inundation and runup tasks, [Macias 2017] is one such example.

40     (a) *The authors completely agree with the referee in that avoiding a minimum depth and using wetting and drying to capture inundation would make the study comparable to field measurements. However, during the study we made a pragmatic choice to represent the islands as voids in the mesh, mainly due to the fact that no large scale topographical data is available in the public domain for any of the islands in the Maldives. Some present studies assume fixed heights of 1.5m to 2m for the islands, however from tsunami observational data as reported in the*
45     *manuscript we find that despite the relative low lying nature of the islands, the relative differences in topographic profiles of the island play a major role in inundation levels across the island. Hence, we decided that due to the lack of data, we would focus on identifying high impact regions at the atoll scale which could be highlighted for further study which could be used to model individual island scale inundations with the availability of high resolution topography and bathymetry. In selecting 0.1m as the minimum depth we also considered the fact that,*
50     *all most all of the inhabited and industrial islands of the Maldives now have sea walls and additional off shore coastal protection which also needs to be taken to account for island scale modelling.Furthermore, we have now specifically stated in the manuscript that inundation is not considered in the study mainly because the topography data is not available and that the results from this study could be used to identify and carryout detailed tsunami assessments of specific islands of interest as part of a future study. These are now included in different parts of the*
55     *manuscript as follows :*

> *Although inundation is not the focus of this work, we found that the introduction of a small minimum depth decreases the computation time of the simulation. [Section 2.3]*

> *Survey results from all of these field studies are included here. However, where the spatial distances between the field observations are very small, we considered the measurement which was closest to the coastline, since the model used here does not capture inundation of islands as explicit simulations of inland flooding would require high-resolution data capturing land features, which is not currently available. [Section 3.2]*

> *Additionally, topographic data, currently unavailable would also provide a means of incorporating inundation modelling, not considered in this study. [Section 4.3]*

**2.3   Response to Minor Comments**

60   1. *Since these are suggestions by the referee to make minor changes to the manuscript to fix figures and typing errors, we have made the changes as suggested in the revised manuscript.*

**2.4 Response to Line by Line Comments**

Has the Thetis model in this set up been validated against traditional tsunami benchmark problems? If not I would suggest taking a look at the problems outlined in [https://nctr.pmel.noaa.gov/benchmark/].

65   1. *The Thetis coastal ocean model has been benchmarked and compared to similar models such as in Pan (2020), where results are presented for some of the standard benchmark cases also outlined in [https://nctr.pmel.noaa.gov/benchmark/]. However, it should be noted that here we used the 2d hydrostatic version of Thetis since we do not focus on inundation and rather on identifying the larger atoll scale patterns of tsunami flow. We appreciate that for more small scale simulations which include very complex topography and inundation processes, we would require the use of the non-hydrostatic*

70   *version of Thetis which has been benchmarked in Pan (2020) marked against standard tsunami inundation cases such as the Okushiri tsunami. In line with the proposed comment by the reviewer we have added the following in the manuscript to highlight this as a future work:*

*The availability of topographic data will also enable higher resolution nearshore simulations using the 3D, non-hydrostatic version of Thetis Pan et al. (2019), which has been validated against standard tsunami benchmarks, at geographical regions of interest identified from this large scale study. [Section 4.3]*

**3   Conclusion**

75   We have attempted to adress the outstanding comments as raised by the reviewer and We hope that these proposed changes are satisfactory to the reviewer, and once again apologies for the oversights in the last round of review.

**References**

Pan, W.: Development of a non-hydrostatic coastal ocean model using the discontinuous Galerkin method, 2020.

Pan, W., Kramer, S. C., and Piggott, M. D.: Multi-layer non-hydrostatic free surface modelling using the discontinuous Galerkin method,
Ocean Modelling, 134, 68 – 83, https://doi.org/https://doi.org/10.1016/j.ocemod.2019.01.003, http://www.sciencedirect.com/science/article/pii/S1463500318302312, 2019.

80